



# Predicting the climate impact of aviation for en-route emissions: The algorithmic climate change function submodel ACCF 1.0 of EMAC 2.53

Feijia Yin[1], Volker Grewe[1,2], Federica Castino[1], Pratik Rao[1], Sigrun Matthes[2], Katrin Dahlmann[2], Simone Dietmüller[2], Christine Frömming[2], Hiroshi Yamashita[2], Patrick Peter[2], Emma Klingaman[3], Keith Shine[3], Benjamin Lührs[4], Florian Linke[4]

[1] Delft University of Technology, Faculty of Aerospace Engineering, 2629HS, Delft, the Netherlands
[2] Deutsches Zentrum für Luft- und Raumfahrt, Institut für Physik der Atmosphäre, 82234 Wessling, Germany
[3] University of Reading, Department of Meteorology, RG6 6AH Reading, United Kingdom
[4] Deutsches Zentrum für Luft- und Raumfahrt, Institut für Lufttransportsysteme, 21079 Hamburg, Germany

*Correspondence to*: Feijia Yin (F.yin@tudelft.nl)

**Abstract** The Modular Earth Submodel System (MESSy) provides an interface to couple submodels to a base model via a modular flexible data management facility. This paper presents the newly developed MESSy submodel, ACCF version 1.0 (ACCF 1.0), based on algorithmic Climate Change Functions version 1.0 (aCCFs 1.0), which describes the climate impact of aviation emissions. The ACCF 1.0 is coupled via the second version of the standard MESSy infrastructure. ACCF 1.0 takes the simulated atmospheric conditions at the location of emission as input to calculate the climate impact (in terms of average temperature response over 20 years (ATR20)) of aviation emissions, including $CO_2$ and non-$CO_2$ impacts, such as from $NO_x$ emissions (via ozone production and methane destruction), water vapour emissions, and contrail-cirrus. The online calculated ATR20 value per emitted mass fuel burn or flown-kilometer using ACCF 1.0 in the ECHAM5/MESSy Atmospheric Chemistry (EMAC) model is presented. We perform quality checks of the ACCF 1.0 outputs in two aspects. Firstly, we compare climatological values calculated by the ACCF 1.0 to previous studies. Secondly, we evaluate the reduction of $NO_x$-induced $O_3$ effects through trajectory optimization, employing the tagging chemistry approach (contribution approach to tag species according to their emission categories and to inherit these tags to other species during the subsequent chemical reactions). Finally, we couple the ACCF 1.0 to the air traffic simulation submodel AirTraf version 2.0 and demonstrate the variability of the flight trajectories when the efficacy of individual effect is considered.

## 1 Introduction

Civil aviation satisfies modern society's mobility needs and is an essential economic driver. Air transportation demand increases at around 4.4% per year and is forecast to maintain that growth for the next decades (Airbus, 2018). Though the





global COVID-19 pandemic has put a tremendous challenge on the aviation industry, aviation (as a fundamental part of the
modern world) will recover eventually[1].

On the other hand, the environmental impact of aviation is increasing at an evenly rapid pace. Aviation contributes 2.5% to
the global anthropogenic $CO_2$ emissions and is responsible for about 3.5% of global warming (Lee et al., 2021). This is because
the non-$CO_2$ effects from aviation in the uppermost troposphere and lowermost stratosphere are as harmful to global climate
change as the $CO_2$ emissions. The non-$CO_2$ effects include ozone ($O_3$) formation and methane ($CH_4$) depletion (causing the
primary mode ozone (PMO) and stratospheric water vapour (SWV) decrease) due to aviation $NO_x$ emissions (Stevenson et al.,
2004; Köhler et al., 2013; Myhre et al., 2007), contrail-cirrus (Heymsfield et al., 2010; Burkhardt and Kärcher, 2011;
Schumann and Graf, 2013; Kärcher, 2018) and their alterations by aerosols direct and indirect effects (Kärcher et al., 2007;
Penner et al., 2009; Myhre et al., 2013; Chen and Gettelman, 2016), and water vapour ($H_2O$) effect (Wilcox et al., 2012). The
non-$CO_2$ effects depend not only on the emission quantity but also on the altitude, geographical location, time, and local
weather conditions (e.g., Frömming et al., 2021). Therefore, it is possible to mitigate aviation's climate impact via operational
measures to avoid climate-sensitive regions associated with non-$CO_2$ effects (Grewe et al., 2017b; Sridhar et al., 2011; Yin et
al., 2018; Matthes et al., 2020).

Information on the climate-sensitive regions, i.e., areas where the non-$CO_2$ effects are significantly enhanced or reduced, is
required to facilitate climate-optimized flight operations. In the earlier research within the EU-project REACT4C[2], Climate
Change Functions (CCFs) were developed. The CCFs are 5D datasets (including longitude, latitude, altitude, time, and
emission type) that describe the specific climate impacts, i.e., the average temperature change in K per flown kilometre or per
emitted mass of the relevant species ($NO_x$ and $H_2O$) locally. The high fidelity CCFs were computed for eight representative
weather situations (five winter patterns and three summer patterns classified by Irvine et al. (2013)) for the North Atlantic
region (Frömming et al., 2021; Grewe et al., 2014a). Grewe et al. (2014a) discussed the development and verification procedure
of CCFs thoroughly. Various application studies have demonstrated the effectiveness of the CCFs in climate-optimized
trajectory calculations (Grewe et al., 2014b; Grewe et al., 2017b). These studies show promising mitigation potential when
using CCFs as inputs for flight trajectory optimization (e.g., a 10% reduction in climate impact for a 1% cost increase). One
of the underlying challenges is that calculating these CCFs is computationally expensive. Thus, with the present computing
performance, it is impossible to use CCFs for real-time calculation, which is necessary for future climate-optimized flight
planning.

To this end, the previous research initiated development (Irvine, 2017; Matthes et al., 2017; van Manen and Grewe, 2019) and
test (Rao et al., 2022) of the so-called algorithmic Climate Change Functions (aCCFs). The aCCFs are algorithmic
approximations of the high fidelity CCFs to represent the correlation of meteorological parameters (e.g., temperature and
geopotential) at the time of emission and the respective average temperature change over a time horizon of 20 years (ATR20).

---

[1]https://www.icao.int/sustainability/Pages/Post-Covid-Forecasts-Scenarios.aspx
[2] www.react4c.eu





Since the aCCFs are essentially mathematical approximations, they can be quickly implemented in Numerical Weather Prediction (NWP) models, thereby serving as a means of advanced meteorological information for flight trajectory planning. The ACCF submodel version 1.0 (ACCF 1.0) of the ECHAM5/MESSy Atmospheric Chemistry (EMAC) model is based on the aCCFs version 1.0 (aCCFs 1.0). The ACCF 1.0 calculates the ATR20 from individual emissions and contrail cirrus effect as a function of the online calculated local weather parameters in EMAC. One can use the ACCF 1.0 in two different ways: 1)

to study the sensitivity of non-$CO_2$ effects (i.e., $NO_x$, $H_2O$, contrail-cirrus) to weather parameters; 2) to couple it with a flight planning tool (e.g., EMAC/AirTraf (Yamashita et al., 2016; Yamashita et al., 2020)) for climate-based routes optimization. This paper elaborates on the modelling approach, the characteristics, and the application of the ACCF 1.0. Please note that, for the first time, we show a consistent set of aCCFs formulas in terms of fuel scenario, metric, and efficacy (aCCFs 1.0). Due to the continuous development of aCCFs, we expect different versions of aCCFs to be released in the future. Accordingly, the

ACCF submodel will be updated.

The structure of the paper follows that section 2 provides a roadmap of the ACCF 1.0 development focusing on different considerations when deriving the first version of contrail aCCFs and the $NO_x$ and $H_2O$ aCCFs. Section 3 presents an overview of the ACCF 1.0, including the model components and the individual aCCFs formulas. The original correlations of the $NO_x$ and $H_2O$ aCCFs were derived in van Manen and Grewe (2019), whereas some coefficients in the equations are updated here

for consistency. Furthermore, the contrail cirrus effect is explained in detail here (and in the supplement). In section 4, we evaluate the performance of the ACCF 1.0 outputs via two types of simulations. First, we compare the climatological aCCFs to other literature studies in terms of their latitudinal and vertical variability. Second, we use the tagging chemistry approach (contribution approach, Grewe et al., 2010 and Grewe et al., 2017a) to evaluate the reduction of $NO_x$-induced $O_3$ effect through climate-optimize flight trajectories based on the $O_3$ aCCF formula. Section 5 implements the ACCF 1.0 with the complete sets

of aCCFs in the AirTraf 2.0 to demonstrate the usage of ACCF 1.0 for climate-optimized flight trajectories. It has to be noted that the two demonstration exercises are academic case studies, which don't intend to suggest an efficient implementation of such climate-optimized trajectories as we present here the extreme case of only considering ecological effects while completely ignoring economic effects in the optimisation (equivalent to a non-combined objective function). One could consider combining the cost and climate objectives in trajectory optimizations to identify eco-efficient flights (e.g., Matthes et al., 2022).

Section 6 discusses further developments of aCCFs before concluding in section 7.

## 2  Roadmap of the MESSy ACCF 1.0 submodel development

The new MESSy submodel ACCF 1.0 consists of a set of aCCFs 1.0, which take relevant local meteorological data as inputs to calculate the ATR20 for a given emission or effect concerning contrails. Figure 1 illustrates the development of the ACCF 1.0, including the previous research on the original CCFs development followed by the aCCFs approach, which is the core of

the ACCF submodel.



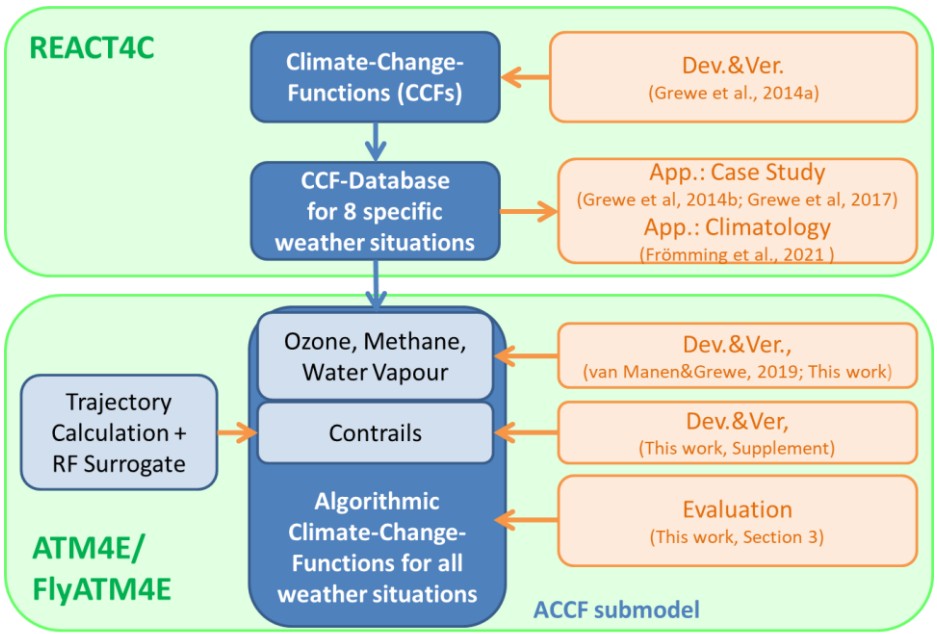

Dev.: Development;    Ver.: Verification;    App.: Application;    RF: Radiative Forcing

**Figure 1 Overview of conceptual development and relevant projects (i.e., REACT4C, ATM4E, and FlyATM4E) leading to the algorithmic Climate Change Functions (aCCFs) and the ACCF submodel.**

The individual CCFs, the basis of the aCCFs, were developed slightly differently. The CCFs of $O_3$, $CH_4$, and $H_2O$ were

calculated using a well-established modelling chain within EMAC (Jöckel et al., 2006; Jöckel et al., 2010). The model follows

a multi-step approach starting with the simulation of the fate of emissions. The impact of pulse emission from a large number

of time-region grid points is efficiently calculated by applying a Lagrangian transport scheme (i.e., following the air parcel).

The radiative forcing (RF) caused by these pulse emissions is computed using the online diagnostic of EMAC radiation scheme.

Grewe et al. (2014a) and Frömming et al. (2021) have described details of this approach.

For the contrail CCF, the Lagrangian trajectories were used to determine the lifetime of a contrail, the temperature, and the

position along the lifetime of a contrail. The Lagrangian trajectories were computed using the ECMWF reanalysis data (ERA-

Interim (Dee et al., 2011)) with winds input to a trajectory model (Methven, 1997). Accordingly, the contrail optical depth and

solar zenith angle were calculated to obtain the contrail RF. The main discrepancy between contrail CCF and the other CCFs

lies in the RF calculation. The contrail RF is calculated using the parametric model described by Schumann et al. (2012),

different from the EMAC radiation scheme. Knowing the RF, to obtain the ATR20 value, the conversion from RF to ATR20

is calculated using the climate response model AirClim (Grewe and Stenke, 2008; Dahlmann et al., 2016) in a consistent way

for all species considered, which differ from the earlier studies.

Based on the CCFs, the regression method was then applied to derive the aCCFs of $O_3$, $CH_4$, $H_2O$ (van Manen and Grewe,

2019), and the contrail cirrus aCCFs (supplement of this paper). The $CO_2$ aCCF is a constant value, which is determined based





on emission scenarios. Note that the values from van Manen and Grewe (2019) and Irvine et al. (2017, supplement to this publication) are updated by the formulas in the present study, as a more consistent conversion to ATR is employed, using slightly different response functions and consistent future scenarios for all species.

## 3 Overview of ACCF 1.0 submodel

### 3.1 Model description EMAC

ACCF 1.0 is a submodel of the global atmospheric-chemistry model EMAC. EMAC is a numerical chemistry-climate-model system that includes submodels describing the tropospheric and middle atmosphere processes and their interaction with oceans, land, and influences from anthropogenic emissions (Jöckel et al., 2010). It uses the second version of the Modular Earth Submodel System (MESSy2 version 2.53; Jöckel et al. 2010) to connect computer codes generated from different institutions. The core atmospheric model is the 5th generation European Center Hamburg general circulation model (ECHAM5 version

5.3.02, Röckner et al. 2006). The model resolution used in the current study is T42L31ECMWF, corresponding to 2.8° by 2.8° in latitude and longitude and 31 vertical hybrid pressure levels up to 10 hPa. The temporal resolution is 12 minutes.

### 3.2 Submodel ACCF 1.0

Figure 2 illustrates the structure of ACCF 1.0 and its interactions with other EMAC submodels. The ACCF 1.0 includes two layers: the Sub-Model Interface Layer (SMIL) and the Sub-Model Core Layer (SMCL). The SMIL manages model

input/output through the CHANNEL submodel (Jöckel et al., 2010). The SMCL is independent of other submodels and contains the code to solve the relevant equations for the individual aCCFs. The input variables to calculate aCCFs in the ACCF submodel are either from the base model calculation (i.e., temperature, geopotential) or from the other EMAC submodels. For instance, the $H_2O$ aCCF is a function of potential vorticity (PV) provided by the submodel TROPOP (Jöckel et al., 2006). The day-time contrail aCCF depends on the Outgoing Longwave Radiation (OLR) at the top of the atmosphere from the submodel

RAD (Dietmüller et al., 2016). The potential contrail coverage (potcov) calculated from the submodel CONTRAIL (Frömming et al., 2014) is used to determine whether persistent contrails can form and may lead to a climate impact by contrails. The supplement of this paper includes a user manual of the submodel ACCF. It describes the namelist settings of the ACCF submodel and includes submodels necessary for coupling input/output variables.



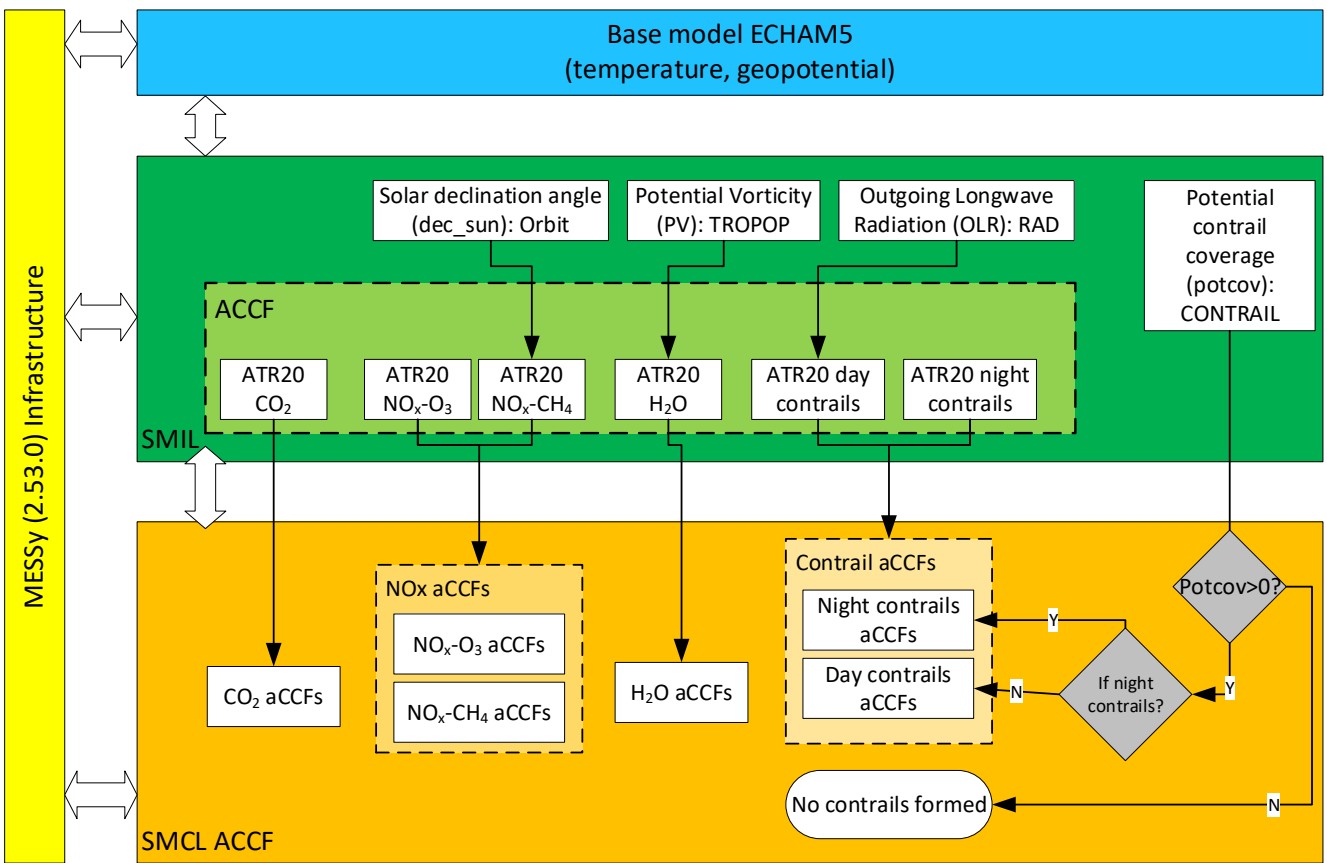

**Figure 2 Overview of EMAC/ACCF submodel structure, the calculation process in the ACCF submodel, and its interaction with the other MESSy submodels. SMIL (submodel interface layer) and SMCL (submodel core layer) are components of MESSy coding standards.**

### 3.3  Basic mechanisms of submodel ACCF 1.0

This section summarizes the formulas of aCCFs 1.0. For full details of the original derivation, the reader is referred to van

Manen and Grewe (2019) and the supplement of this paper. The complete set of the aCCFs 1.0 computes the ATR20 of $CO_2$

emissions, $H_2O$ emissions, $NO_x$ emissions (forming $O_3$ and decreasing $CH_4 + PMO$), and day/night contrail-cirrus.

### 3.3.1    Synoptic on a selected day

The individual non-$CO_2$ aCCFs depend on weather parameters, e.g., temperature, geopotential, and potential vorticity. A one-

day simulation on December 18[th] 2015 was performed to demonstrate such correlations. Figure 3 shows the geographical

distribution of a) temperature, b) potential vorticity, and c) geopotential over Europe at the pressure level of 250 hPa on the

same day. These parameters are calculated by running the EMAC model nudged towards the ERA-interim data and will be

used to calculate the non-$CO_2$ aCCFs (see the following sections).





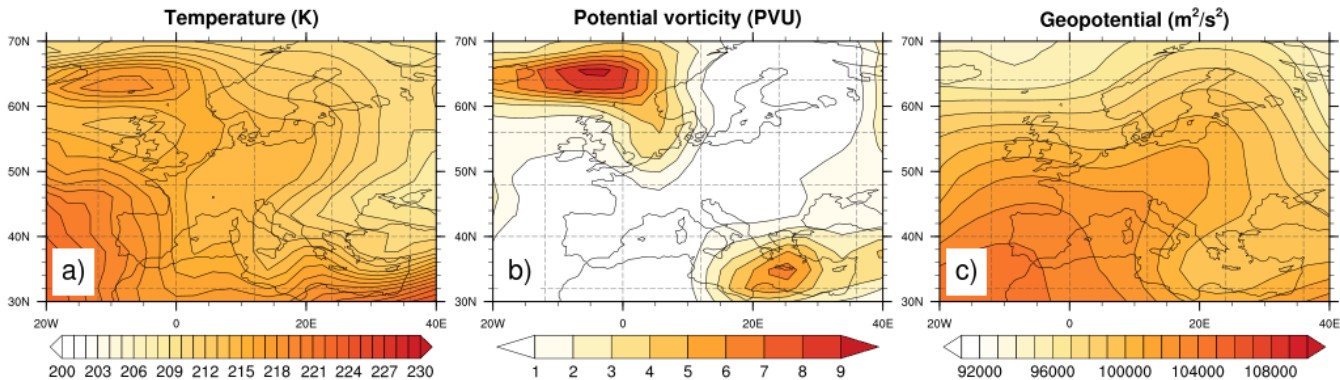

**Figure 3 Geographical distribution of a) Temperature (K); b) Potential Vorticity in standard potential vorticity unit (PVU, 1 PVU=$10^{-6}$ K m$^2$ kg$^{-1}$ s$^{-1}$); c) Geopotential (m$^2$/s$^2$) over Europe at 250 hPa on December 18$^{th}$ 2015.**

### 3.3.2 CO$_2$ aCCF

CO$_2$ is a long-lived species, and hence, the climate impact of aviation's CO$_2$ depends only on the amount of CO$_2$ emitted. Therefore, the CO$_2$ aCCF is calculated using the nonlinear climate–chemistry response model AirClim, assuming a 1 Tg fuel use in 2017. The CO$_2$ aCCF then represents the average temperature response of CO$_2$ for 2017-2036 in K/kg(fuel) (named P-ATR20$_{CO2}$). As a result, a constant value of 7.48e-16 K/kg(fuel) was obtained. For the same amount of emission in 2017, but with an annual growth rate according to a Business As Usual (BAU) future scenario as given by Grewe et al. (2021), the ATR20 for CO$_2$ (named F-ATR20$_{CO2}$) was 7.03e-15 K/kg(fuel). A conversion factor of 9.4 was derived from the P-ATR20$_{CO2}$ to F-ATR20$_{CO2}$.

### 3.3.3 NO$_x$ induced aCCFs

The aviation NO$_x$ emission (NO$_x$=NO+NO$_2$) leads to O$_3$ formation via a catalytic reaction. NO reacts with HO$_2$ forming NO$_2$. Due to photodissociation, NO$_2$ forms O($^3$P), leading to the O$_3$ formation. The O$_3$ formation, on the other hand, enhances the OH production (e.g., Grewe et al., 2017a), hence causing a shift of the OH/HO$_2$ ratio towards OH. The additionally formed OH leads to the oxidation of CH$_4$.

Furthermore, the destruction of CH$_4$ leads to a reduced O$_3$ production rate as feedback to the O$_3$ concentration. This O$_3$ change is called primary mode ozone (PMO) (Wild et al., 2001). The effect of PMO is much smaller than the initial O$_3$ production. However, PMO has a longer lifetime (is bound to the CH$_4$ perturbation) than the initial O$_3$ production. Furthermore, because of the CH$_4$ oxidation, less CH$_4$ enters the stratosphere, which again reduces the SWV. Since H$_2$O is a greenhouse gas, the decrease of SWV reduces the warming effect of H$_2$O (Myhre et al., 2007). The overall aviation-induced NO$_x$ effects include the short-term O$_3$ increase and long-term CH$_4$ reduction (also the CH$_4$-related PMO and SWV decrease). The current NO$_x$ aCCF addresses the impact of short-term O$_3$ production and CH$_4$ destruction, and PMO reduction. SWV decrease is not taken into account because of its low magnitude. The corresponding formulas are presented below.





*$NO_x$-induced $O_3$-aCCF*

The earlier research showed the impact of weather patterns and related transport processes on the contribution of aviation $NO_x$

emissions to $O_3$ and $CH_4$ concentrations (Grewe et al., 2017c, Frömming et al., 2021; Rosanka et al., 2020). For instance, Grewe et al. (2017c) and Frömming et al. (2021) showed that a unit $NO_x$ emission within a high-pressure blocking situation leads to more $O_3$-induced RF than a $NO_x$ emission west of this high pressure area because the transportation pathways differ significantly. Air parcels starting within the high-pressure system are transported to the tropics and lower altitudes, experiencing a more active chemical regime and faster $O_3$ production (Rosanka et al., 2020).

The analysis by van Manen and Grewe (2019) independently looked at correlations of CCFs data describing the atmospheric state (meteorological and chemical data) at the time of emission. They found the best correlation representing the impact of ozone changes caused by a local $NO_x$ emission with the geopotential and temperature. This indicates that the weather regime at the time of emission essentially controls the air parcel's fate in which $NO_x$ is emitted. Thereby, the $O_3$-aCCF in K/kg($NO_2$) is developed based on temperature (T) in K and geopotential ($\Phi$) in $m^2/s^2$. For an atmospheric location $(x, y, z)$ at time t with

$T = T(x, y, z, t)$ and $\Phi = \Phi(x, y, z, t)$, the $O_3$-aCCF can be found in Eq. (1).

$$aCCF_{O_3}(T, \Phi) = -2.64 \times 10^{-11} + 1.17 \times 10^{-13} \times T + 2.46 \times 10^{-16} \times \Phi - 1.04 \times 10^{-18} \times T \times \Phi$$

$$aCCF_{O_3}(T, \Phi) = \begin{cases} aCCF_{O_3}(T, \Phi) & for \quad aCCF_{O_3} > 0 \\ 0 & else \end{cases} \tag{1}$$

$$aCCF_{O_3} \approx P\text{-}ATR20_{O_3}$$

where P-ATR20$_{O3}$ is the ATR20 for a pulse emission.

Figure 4 a) shows an example of the $O_3$-aCCF in [K/kg($NO_2$)] on December 18$^{th}$ 2015 over Europe at 250 hPa. The contour lines indicate the geopotential, and it is noticeable that the $O_3$-aCCF strongly follows the geopotential distribution. Overall,

the changes in $O_3$ concentration caused by $NO_x$ emissions have warming effects.

*$NO_x$-induced $CH_4$-aCCF*

The analysis by van Manen and Grewe (2019) showed the highest correlation of the $CH_4$ response to $NO_x$ emissions with geopotential and the mean incoming solar radiation, i.e., combining the initial transportation pathway with an indicator for both seasons and available incoming radiation. Therefore, the $CH_4$-aCCF in K/kg ($NO_2$) is based on geopotential ($\Phi$) in $m^2/s^2$

and incoming solar radiation at the top of the atmosphere as a maximum value over longitude ($F_{in}$) in $W/m^2$. For an atmospheric location $(x, y, z)$ at time t with $\Phi = \Phi(x, y, z, t)$, the $CH_4$-aCCF can be found in Eq. (2).

$$aCCF_{CH_4}(\Phi, F_{in}) = -4.84 \times 10^{-13} + 9.79 \times 10^{-19} \times \Phi - 3.11 \times 10^{-16} \times F_{in} + 3.01 \times 10^{-21} \times \Phi \times F_{in}$$

$$aCCF_{CH_4}(\Phi, F_{in}) = \begin{cases} aCCF_{CH_4}(\Phi, F_{in}) & for \quad aCCF_{CH_4} < 0 \\ 0 & else \end{cases} \tag{2}$$

$$aCCF_{CH_4} \approx P\text{-}ATR20_{CH_4}$$

where P-ATR20$_{CH4}$ represents the ATR20 for pulse emission, and $F_{in}$ is calculated by Eq. (3)





$$F_{in} = S \times \cos\theta, \text{ with } S = 1360 \ W/m^2$$
$$\cos\theta = \sin\varphi \times \sin d + \cos\varphi \times \cos d, \text{ and} \qquad (3)$$
$$d = -23.44° \times \cos(360/365 \times (N+10))$$

where S is the total solar irradiance, θ is the solar zenith angle, φ is latitude, and d is the declination angle, defined by the time

of year via the day of the year N.

*NOx-induced PMO-aCCF*

The effects of PMO and SWV decrease are not included in Eq. (2) but might be simply regarded as an offset of the $CH_4$-aCCF

with a linear scaling factor (e.g., Skowron et al., 2013), as they are primarily driven by the $CH_4$ change. Here we apply a

constant factor of 0.29 to the $CH_4$-aCCF calculated in Eq. (2) to account for the PMO effect (Dahlmann et al., 2016). The

PMO-aCCF is then described by Eq. (4).

$$aCCF_{PMO} = 0.29 \times aCCF_{CH_4}$$
$$\qquad (4)$$
$$aCCF_{PMO} \approx P\text{-}ATR20_{PMO}$$

Figure 4 b) shows an example of the combined $CH_4$-CCF and PMO-aCCF in $K/kg(NO_2)$ on December 18th 2015 over Europe

at 250 hPa. The overlaid contour lines represent the geopotential on the same pressure level and time step. We can see that the

decrease in $CH_4$ concentration caused by $NO_x$ emissions has cooling effects. Here, the cooling effects are overcompensated by

the warming effects of $O_3$. The overall effects of $NO_x$ emissions are expected to be warming, as seen in Figure 4 c), which

shows the summation of $O_3$-aCCF, $CH_4$-aCCF, and PMO-aCCF.

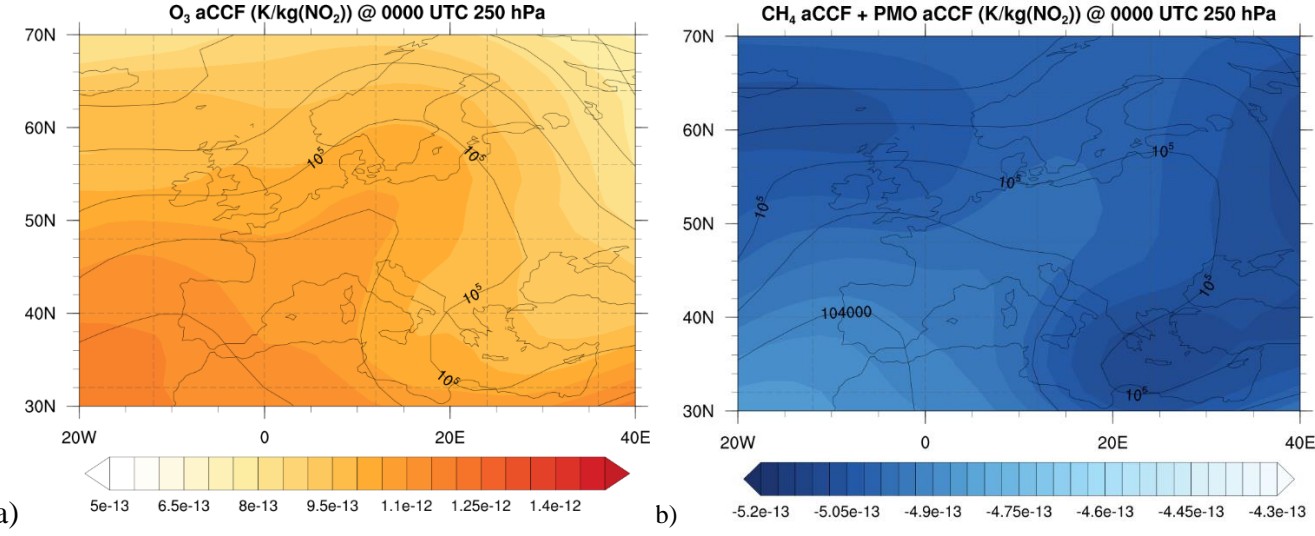





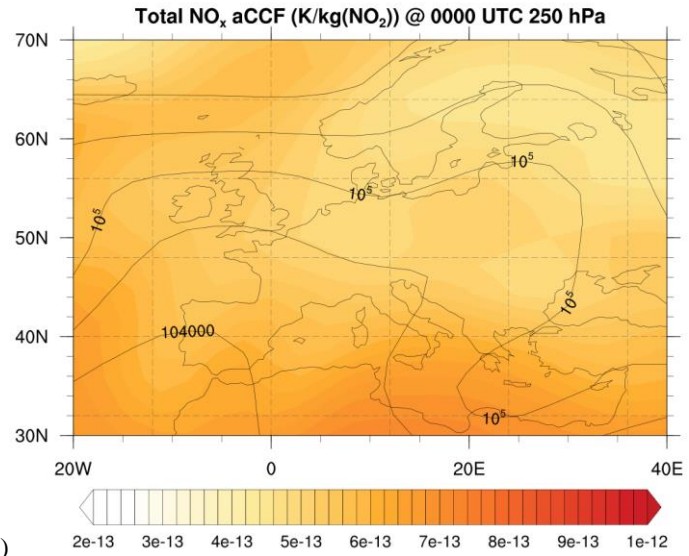

c)

**Figure 4 NOₓ aCCF in K/kg(NO₂) on December 18ᵗʰ 2015 at 250 hPa for: a) O₃-aCCF; b) the combined CH₄-aCCF and PMO-aCCF and c) the total NOₓ aCCF (O₃-aCCF + CH₄-aCCF + PMO-aCCF). The black contour lines are geopotential in m²/s².**

### 3.3.4 H₂O-aCCF

The $H_2O$ emission's climate impact largely depends on its residence time. The likelihood of removing (rain-out) the emitted $H_2O$ decreases with altitude up to the tropopause. Or vice versa, the $H_2O$ emission's residence time increases with the height and shows a sharp gradient at the tropopause (Grewe and Stenke, 2008; Wilcox et al., 2012). Hence the distance to the tropopause is already a good indicator of the $H_2O$'s lifetime. There are different tropopause definitions, for instance, temperature lapse rate (including the World Meteorological Organization (WMO), thermal tropopause (WMO, 1957)) and potential vorticity (PV) (Kunz et al., 2011). The WMO thermal tropopause and the PV dynamical tropopause may differ locally (Grewe and Dameris, 1996). van Manen and Grewe (2019) showed that PV is a better indicator for the $H_2O$-aCCF, since PV can also be used as a definition between tropospheric and stratospheric air masses.

The $H_2O$-aCCF in K/kg(fuel) is based on PV in the standard potential vorticity unit (PVU). For an atmospheric location $(x, y, z)$ at time t with $PV = PV(x, y, z, t)$, the $H_2O$-aCCF can be found in Eq. (5).

$$aCCF_{H_2O}(PV) = 2.11 \times 10^{-16} + 7.70 \times 10^{-17} \times |PV|$$
$$aCCF_{H_2O} \approx P\text{-}ATR20_{H_2O}$$

(5)

Figure 5 shows an example of the $H_2O$ aCCF in K/kg(fuel) on December 18ᵗʰ 2015 over Europe at 250 hPa. One can notice that the $H_2O$ has warming effects in general, and the highest values occur at the location where the potential vorticity is also high (see Figure 3 b)).





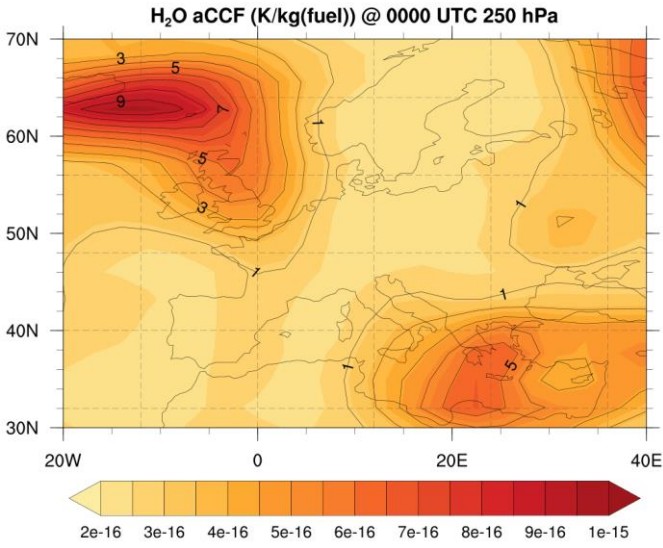

**Figure 5 H₂O aCCF (colored contour) in K/kg(fuel) and potential vorticity (black contour) in standard potential vorticity unit (PVU) on December 18ᵗʰ 2015, at 250 hPa.**

### 3.3.5    Contrail cirrus aCCF

Contrail cirrus is short-lived. Because of its contrasting effects on shortwave and longwave radiation, contrail cirrus's radiative and climate effects distinguish between day- and night-time. Thus, the contrail cirrus aCCF in K/km has been developed for

the day- and night-time conditions by E. Irvine (now Klingaman) based on reanalysis data (Klingaman and Shine, supplement). Unlike the other aCCFs formulas in calculating the P-ATR20 value directly, the algorithm of contrail cirrus estimates the global and annual mean RF using the parametric equation of Schumann et al. (2012). Accordingly, the contrail-cirrus aCCF (an approximation of ATR20) for pulse emissions ($P\text{-}ATR20_{contrail}$) is obtained as a product of the RF value and a constant of 0.0151 K/W/m² derived using AirClim model.

*Night-time contrails aCCF*

Night-time contrails refer to contrails with their entire (6 hours in this paper) lifetime occurring at night. Since these contrails exist only during hours of darkness, they cause only longwave RF, so their net RF must be positive (warming). The scatterplot of relevant meteorological variables against net RF of night contrails was used to identify which parameters had the strongest relationships with the net RF (see Klingaman and Shine supplement). It was found that the local temperature can provide

reasonable approximations for the night contrails' radiative effects. By using the nonlinear regression method, the RF of night-time contrails ($RF_{contrails-night}$) in W/m² is derived based on temperature (T) in K. For an atmospheric location $(x, y, z)$ at time t, with $T = T(x, y, z, t)$, the RF of night-time contrail-cirrus can be found in Eq. (6). Please note that correlation is not valid for temperatures less than 201K. For temperatures below 201K, the value should be set to 0.





$$RF_{contrails\text{-}night} = \begin{cases} 10^{-10}\times(0.0073\times10^{0.0107\times T}-1.03) & for \quad T>201K \\ 0 & else \end{cases} \tag{6}$$

By multiplying the factor of 0.0151, the night-time contrails aCCF in K/km is calculated in Eq. (7).

$$aCCF_{contrails-night} = RF_{contrails-night}\times0.0151$$
$$aCCF_{contrails\text{-}night} \approx P\text{-}ATR20_{contrails-night} \tag{7}$$

*Day-time contrails aCCF*

Day-time contrails refer to contrails that form and dissipate during daylight or have a part of their 6-hour lifetime during the day. The RF of day-time contrails ($RF_{contrails-day}$) in W/m$^2$ is based on the OLR in W/m$^2$ at the top of the atmosphere at the
time and location of the contrail formation. Therefore, for an atmospheric location $(x, y)$ at time t with $OLR(x, y, t)$, the RF of day-time contrail-cirrus can be found in Eq. (8). Please note that Eq. (8) will predict negative RF for OLR < -193 W/m$^2$ and positive RF for any larger OLR values.

$$RF_{contrails-day} = 10^{-10}\times(-1.7-0.0088\times OLR) \tag{8}$$

Similarly, the day-time contrails aCCF in K/km is calculated in Eq. (9).

$$aCCF_{contrails-day} = RF_{contrails-day}\times0.0151$$
$$aCCF_{contrails-day} \approx P\text{-}ATR20_{contrails-day} \tag{9}$$

Please note that in the ACCF submodel, the contrail aCCF is only activated when the potential contrail coverage is larger than zero. Depending on the time of the contrail-cirrus occurring, either day- or night-contrail-cirrus aCCF calculation is used. Figure 6 shows an example of the day- and night-time contrail aCCF on December 18$^{th}$ 2015 over Europe at 250 hPa: a) 1200 UTC and b) 0000 UTC. One can see that the contrail aCCF depends on the formation time. For instance, at the exact location
(e.g., over Ireland), contrail formed at 1200 UTC has a cooling effect, whereas at 0000 UTC has a warming impact.





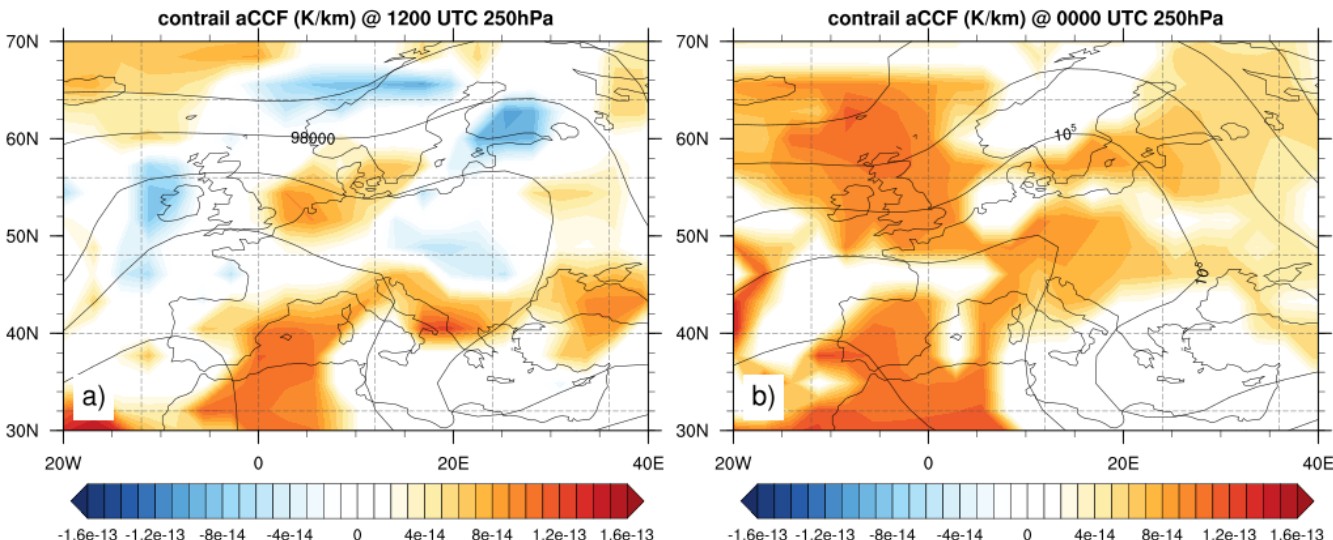

**Figure 6 Contrail-cirrus aCCFs (colored contour) in K/km and geopotential height (black contour) in $m^2/s^2$ on December 18th 2015 at 250 hPa: a) 1200 UTC; b) 0000 UTC.**

### 3.4 Physical climate metric and efficacy applied in the ACCF submodel

The aCCFs formulas provided in section 3.3 calculate the climate impact of $O_3$, $CH_4$, PMO, $H_2O$, and contrail-cirrus consistently in P-ATR20, i.e., for a pulse emission. With the pulse emission, one could compare, for instance, the future impact of emissions in a given year. When a non-pulse emission is considered, e.g., an increased emission scenario representing the growth of air traffic, the metrics of pulse emission can be converted (Fuglestvedt et al., 2010).

Here we demonstrate an example of converting the P-ATR20 to the ATR20 of the future BAU emission scenario (F-ATR20)
derived by Grewe et al. (2021). We determined the climate metrics conversion factors for the aCCFs of $O_3$, $CH_4$, PMO, $H_2O$, and contrail cirrus using the AirClim model. We performed two simulations with pulse emissions in 2017 and future emission scenario BAU, respectively. For both simulations, we calculate the factor between ATR20 and RF for each effect and use the ratio between these values as conversion factors. Table 1 shows the conversion factors from the P-ATR20 to the F-ATR20 metric. In the namelist of the ACCF 1.0, these metric conversion factors can be changed depending on the chosen scenario for
different purposes (see supplement).

The efficacy of the individual forcing agents ($O_3$, $CH_4$, PMO, $H_2O$, and contrail-cirrus), which consider the different effects of these forcing agents in producing global temperature change (e.g., Hansen et al., 2005), are not included in the aCCF formulas in section 3.3. However, they can be easily included via namelist settings of the ACCF submodel (see the user manual in the supplement for namelist settings). The present study implemented the forcing efficacies in Lee et al. (2021), as shown in Table
1. The final output of the ACCF submodel is a product of the output of aCCFs formulas in section 3.3, the metric conversion factor, and the efficacies.





**Table 1 Example values of climate metrics conversion factors from ATR20 of a pulse emission in 2017 (P-ATR20) to ATR20 of future BAU emission scenario (F-ATR20) and efficacies of different species/contrail-cirrus effect. The efficacies are taken from Lee et al. (2021).**

| Descriptions | Metric conversion factors (P-ATR20 →F-ATR20) | Efficacy |
|---|---|---|
| $CO_2$ | 9.4 | 1.0 |
| $NO_x$-$O_3$ | 14.5 | 1.37 |
| $NO_x$-$CH_4$ | 10.8 | 1.18 |
| $NO_x$-$CH_4$-PMO | 10.8 | 1.18 |
| $H_2O$ | 14.5 | 1.0 |
| Contrail-cirrus | 13.6 | 0.42 |

## 4 ACCF model simulations

In this section, we present the application of the submodel ACCF, how it describes the climate effects of aviation emissions and how it can be used for aircraft trajectory optimisation. This section also presents the quality check of ACCF submodel outputs. Firstly, we compare the climatology of the prototype aCCFs for $O_3$, $CH_4$, $H_2O$, and contrail-cirrus to results from the literature. Secondly, we study the $O_3$ RF change caused by the air traffic emissions through the AirTraf submodel calculated online for cost- and climate- optimal flights, respectively. The climate-optimized flights minimized the $NO_x$-induced $O_3$ effect computed using Eq. (1).

### 4.1 Climatology of aCCFs

The climatological aCCFs are calculated for all meteorological situations emerging over a one-year nudged simulation in 2016. The climate metric conversion factors and the efficacies in Table 1 are considered. Figure 7 a)-c) shows the annual and zonal mean aCCF from $O_3$, $CH_4$ combined with PMO, and total $NO_x$ ($O_3$+$CH_4$+PMO), respectively. The considered region is over the northern hemisphere and between 150-300 hPa.

The warming effects of $O_3$ increase with the altitude and towards the lower latitudes, which is in line with other studies. For instance, Fig A.2 of Dahlmann et al. (2016) shows that the global annual mean RF of aviation $NO_x$-induced $O_3$ increases with the pressure altitude. Figure 8 of Grewe and Stenke (2008) shows the global mean temperature change of $NO_x$-induced $O_3$ for 2100, considering a constant emission from 2050-2100. Due to different emission scenarios, the absolute value in Fig. 7 of this study is much lower (order of magnitudes). However, when comparing the vertical and lateral variability in the vertical range of 150 and 300 hPa (typical flight corridor range), a similar pattern can be observed.

In comparison, the cooling effect of $CH_4$ (including PMO) increases towards lower altitudes but shows less dependency on latitude than $O_3$ at the lower altitude. That is to say, if the flight altitude is reduced, one would expect more substantial cooling effects due to $NO_x$-induced $CH_4$ depletion. Such phenomena are in line with the study of Frömming et al. (2012), where it was shown that the $CH_4$ mean RF reduces when flying lower. Furthermore, when comparing Fig. 8 of Grewe and Stenke (2008)



and Fig. A.2 of Dahlmann et al. (2016) in the same vertical range, we notice some discrepancies in the $CH_4$ aCCF pattern in the latitudinal directions. Both Fig. 8 and Fig. A.2 show that the cooling effects of $CH_4$ increase towards lower latitudes. This was also observed in Köhler et al. (2013). However, Fig. 7 b) shows an opposite trend, which needs further diagnosis in future

studies. Since the value of $CH_4$ aCCF is about five times smaller than the $O_3$ aCCF, one can consider the mismatch of $CH_4$ aCCF to be of minor importance.

Figure 7 d) shows the annual zonal mean $H_2O$ aCCF. The warming effects of $H_2O$ increase with altitude and towards the polar region, which matches well with the previous study of Grewe and Stenke (2008), confirming that ACCF accurately represents the variations in global climate change of aviation $H_2O$ emissions at the different regional locations and different altitudes.




**Figure 7 Annual zonal mean aCCFs (F-ATR20) in the northern hemisphere and the between 150 to 300 hPa attributed to a) NOx-O₃ effects; b) NOx-CH₄ (+PMO) effects; c) overall NOx effects (O₃+CH₄+PMO); d) H₂O effects**

Figure 8 shows the zonal mean climatological value of contrail cirrus aCCFs in K/km by combining the day and night effects.
The RF and hence the F-ATR20 is calculated at the location where contrails could be formed. We compare the climatological

contrail-cirrus aCCF with the values presented in the previous literature (Fig. A.2. of Dahlmann et al. (2016), where the annual
zonal mean contrails RF per flown km are calculated using normalized emissions). We notice that the order of magnitude and
the profile of the contrail aCCF matches the study of Dahlmann et al. (2016).

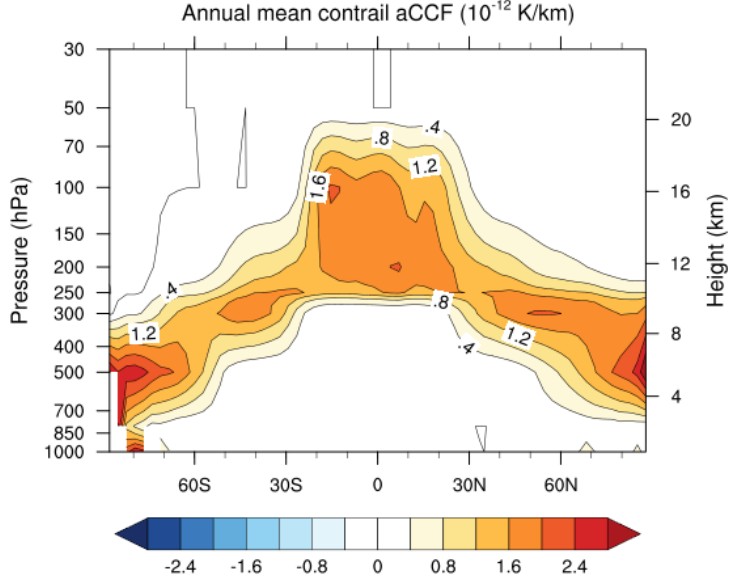

**Figure 8 Annual zonal mean contrail aCCF (F-ATR20) in K/km: combined effects of day and night contrails.**

### 4.2 Radiative forcing calculation of aircraft emissions using EMAC submodels

To demonstrate the usage of the ACCF 1.0 in aircraft trajectory optimisation considering non-$CO_2$ climate effects, we use the
$O_3$ aCCF to calculate the RF due to aviation $NO_x$-induced $O_3$ by combining ACCF with AirTraf, TAGGING, and RAD (EMAC
submodels). Figure 9 shows how an aircraft trajectory from the departure to the arrival airport is guided through climate-

sensitive regions, described with the help of the ACCF submodel. Providing atmospheric perturbations in reactive species to
the TAGGING submodel calculates associated ozone changes, and eventually, radiative impacts are characterized in the RAD
submodel. For the demonstration, we optimize the flight trajectories of a subset of daily European flights concerning either
minimum cost (simple operating cost option in the AirTraf submodel (Yamashita et al., 2020)) or minimum climate impact
from only $NO_x$-induced $O_3$ effect only. In two different simulations, the associated $NO_x$ emissions alter $O_3$ concentrations and

thus their RF differently. The hypothesis of the reduced RF in climate-optimized routes would prove the concept of the $O_3$
aCCFs. For a more detailed study, the climate impact of aviation $NO_x$ emissions should be a combination of $O_3$, $CH_4$, PMO,
and SWV decrease. Here we focus on the short-term $O_3$ effect to better understand the particular feature of the $O_3$ aCCF.



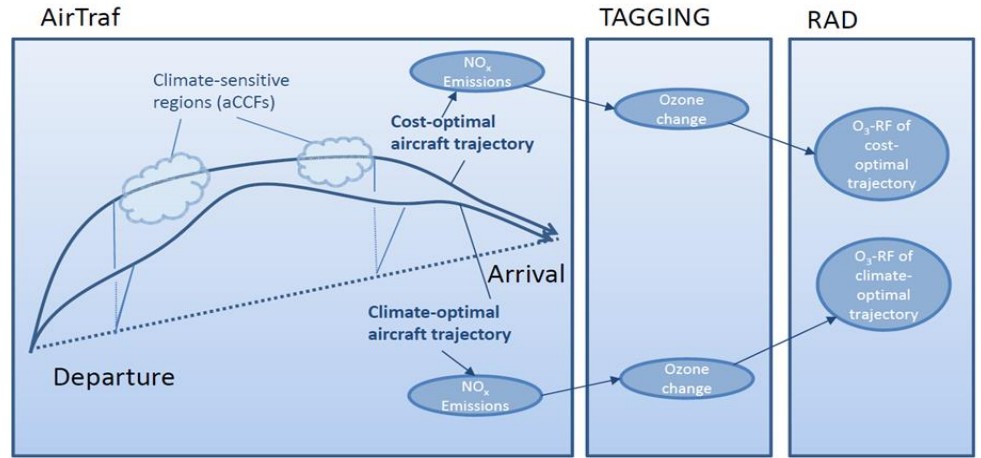

**Figure 9 Sketch of the radiative forcing calculations for ozone changes caused by online air traffic NO$_x$ emissions for cost- and climate-optimized flight trajectories.**

In line with the simulation scheme above, we configured the EMAC model with a list of EMAC submodels. In addition to the standard submodels, we use AirTraf 2.0 (Yamashita et al., 2020) to calculate the air traffic emissions from different flight trajectories, MECCA (Module Efficiently Calculating the Chemistry of the Atmosphere, Sander et al., 2005) and SCAV (SCAVenging, Tost et al., 2006a) to represent the chemical kinetics of EMAC. We also use TAGGING 1.0 (Grewe et al.,

2017a) to tag the contributions of emissions to concentrations. The radiation flux change of the NO$_x$-induced O$_3$ change is calculated using the submodel RAD (Dietmüller et al., 2016). The complete list of used EMAC submodels in this simulation can be found in Table A.1 of the Appendix.

    The simulation setup for trajectory optimization is given in Table 2. 85 daily European flights are used. The constant flight Mach number 0.82 combined with the wind speed will result in different ground speeds. For cost-optimized flight trajectories,

simple operating cost calculated using Eq. (10) is the objective function. For climate-optimized flight trajectories, the F-ATR20 of NO$_x$-induced O$_3$ is used as the objective function. There are 11 design variables to express a flight trajectory. Five variables control the vertical change between flight levels of 29000 ft (FL290) and 41000 ft (FL410), and six variables control the lateral shift. The Adaptive Range Multi-objective Genetic Algorithm (ARMOGA version 1.2.0, (Sasaki and Obayashi, 2005; Sasaki et al., 2002)) is implemented for trajectory optimization.

$$\text{cost} = C_t \cdot t + C_f \cdot m_{fuel} \tag{10}$$

where $t$ is the flight time in hrs, $m_{fuel}$ is the fuel consumption in kg, $C_t$ is the flight time related cost in €/hr; and $C_f$ is the fuel related cost in €/kg (fuel).


**Table 2 AirTraf simulation setup for trajectory optimizations considering cost minimum and climate minimum (only NOₓ-O₃ effect), respectively.**

| Description | | |
|---|---|---|
| AirTraf option | Cost-optimized | Climate-optimized |
| ECHAM5 Resolution | T42/L31ECMWF (2.8°× 2.8° in latitude and longitude, 31 vertical pressure levels up to 10 hPa, a time step of 12 minutes) | |
| Flight plan | 85 daily European flights | |
| Aircraft / Engine type | A330/CF6 engine model | |
| Flight altitude in feet | [FL290, FL410] | |
| Optimization objective | Minimum simple operating cost | Minimum F-ATR20 of NOₓ-O₃ |

Figure 10 shows the calculated flight trajectories on a single day for the minimal cost (red) and the minimal $NO_x$-$O_3$ climate impact (green). Figure 10 a) shows the changes in flight altitudes, and Figure 10 b) shows the lateral shifts of flight trajectories

aggregated along the vertical direction. For cost-optimized flights, the aircraft tends to fly as high as possible within the vertical constraints to maximize aerodynamic efficiency, reducing fuel consumption and the associated operational cost. As for the climate-optimized routine, the situation is much more complicated. The climate impact of $O_3$ attributed to the $NO_x$ emissions depends on multi-criteria, e.g., the emitted quantity, time, location, and weather. On average, the altitudes of climate-optimized flights are lower than those of the cost-optimized flights. We also notice from Figure 10 b) that some flights tend to shift

northward to reduce the $NO_x$-$O_3$ climate impact.

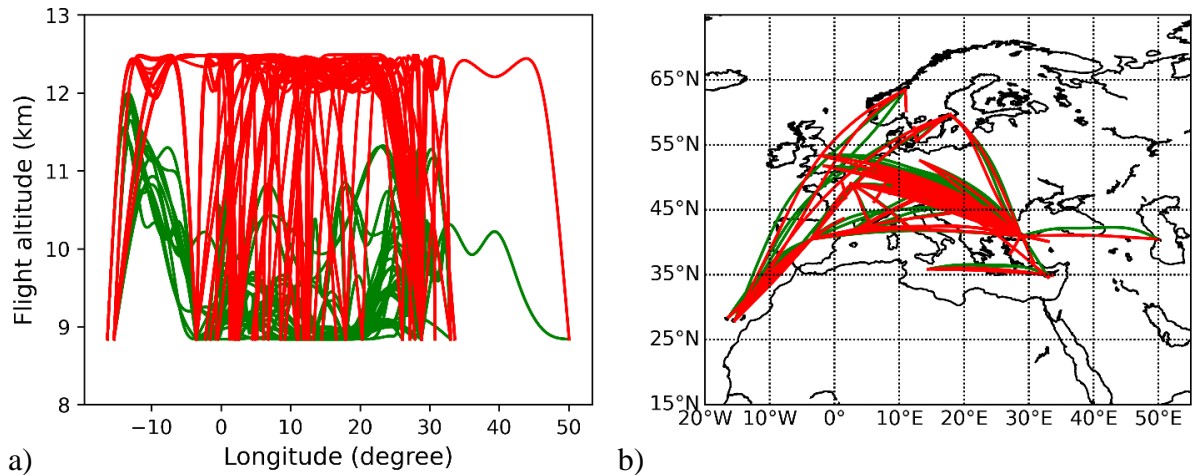

a)                                                b)

**Figure 10 Calculated daily flight trajectories in a) vertical variation and b) lateral variation using AirTraf for cost-optimized (red) and climate optimal considering only the NOₓ-O₃ effects (green).**

The flight characteristics and performance data are summarized in Table 3. Compared to the cost-optimized flights, the fuel

consumption of the climate-optimized flights is 11% higher, and the $NO_x$ emissions are 15% higher. The total cost of climate-optimized flights is about 5% higher than that of the cost-optimized flights.



**Table 3 Daily sum over the flight-plan of the characteristics of the cost-optimized and the NOₓ-O₃-optimized flights.**

| Parameters | Cost-optimized | NOₓ-O₃-optimized | Diff % |
|---|---|---|---|
| Fuel consumption [Tons] | 728 | 810 | +11 |
| NOₓ emissions [Tons] | 7.26 | 8.33 | +15 |
| Flight time [hrs] | 157 | 156 | -0.6 |
| Flight distance [km] | 134000 | 134346 | +0.3 |
| Cost [thousand EUROs] | 636.56 | 667.76 | +4.9 |

Having the flight trajectories and their respective performance calculated with AirTraf using ACCF values (Figure 10), $NO_x$ emissions from cost- and climate-optimal trajectories are then integrated into the global EMAC model as tagged species by the EMAC/TAGGING submodel. This allows identifying the contributions of different $NO_x$ emission sources to the atmospheric changes of the $NO_x$ and $O_3$ concentrations. This showcase simulation using tagging chemistry was run for three months, from January to March 2016. Figure 11 shows relative changes in monthly mean mixing ratio distribution of a) $NO_x$ in mol/mol and b) $O_3$ in mol/mol, comparing effects caused by $NO_x$ emissions from climate-optimized flight trajectories with the effect of cost-optimized trajectories (baseline) in March 2016. The figure is presented in the vertical cross-section. The climate-optimized trajectories emit $NO_x$ at a lower altitude than the cost-optimized trajectories; therefore, we see an increase in the $NO_x$ mixing ratio at the lower altitude (indicated by the red color in Figure 11 a)). As a result, the $O_3$ production is shifted downwards (see Figure 11 b)). The residence time of $O_3$ at the lower altitude is shorter due to a more efficient wash-out. Therefore, the calculated RF of the $NO_x$ induced $O_3$ for the climate-optimized flights (13.3 mW/m$^2$) is about 2% less than that of the cost-optimized flights, which confirms that the climate-optimized flight trajectories based on the $O_3$ aCCF reduce the associated $NO_x$-$O_3$ climate effect.

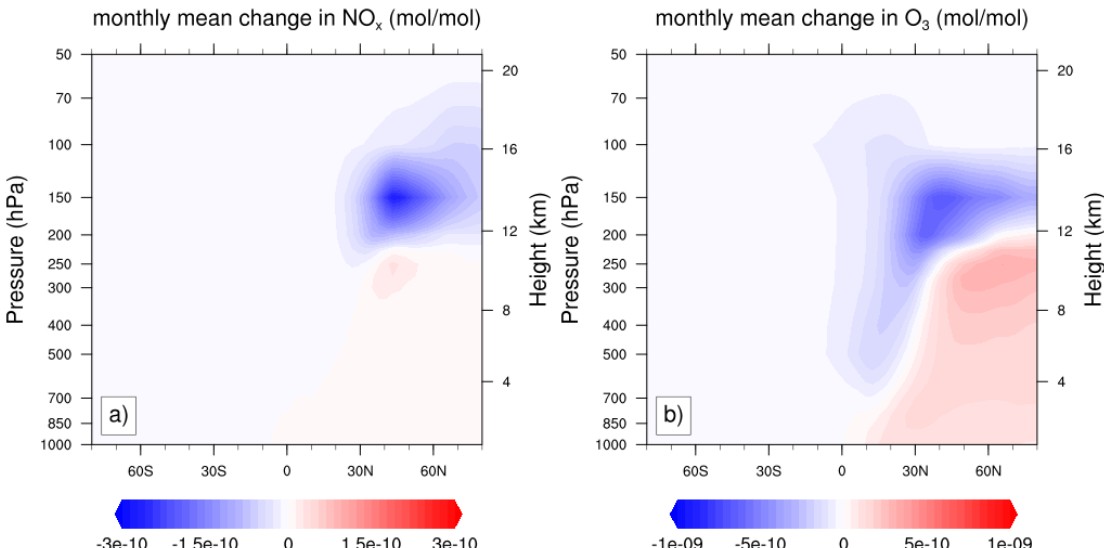

**Figure 11 Changes of a) NOₓ mixing ratio and b) resulting changes in O₃ mixing ratio caused by NOₓ-O₃-optimized flight trajectories using only O₃ aCCF. The baseline is cost-optimized flights.**

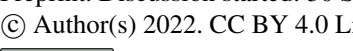

## 5 Application of the ACCF submodel for trajectory optimization

This section demonstrates the application of the ACCF submodel to assess the aviation climate effects during trajectory optimization. In the previous research, Yamashita et al. (2020) implemented the ACCF submodel in AirTraf 2.0 to obtain climate-optimized trajectories. Here, we update the ACCF submodel outputs using the physical climate metric F-ATR20 and consider the efficacy of radiative effects.

### 5.1 Simulation setup

We couple the ACCF 1.0 with the AirTraf 2.0 in this simulation. In the AirTraf 2.0, two optimization objectives are considered, respectively: cost- and climate-optimized. The simulation setup can be seen in Table 4. In this section, the climate-optimized trajectory minimizes the total F-ATR20 of $CO_2$, $NO_x$ (summation of $O_3$, $CH_4$, and PMO), $H_2O$, and day/night contrail-cirrus, including the efficacies of individual species/contrail-cirrus as shown in Table 1.

**Table 4 AirTraf simulation setup for trajectory optimizations considering cost minimum and climate minimum.**

| Description | | |
|---|---|---|
| AirTraf option | Cost-optimized | Climate-optimized |
| ECHAM5 Resolution | T42/L31ECMWF (2.8°× 2.8° in latitude and longitude, 31 vertical pressure levels up to 10 hPa, a time step of 12 minutes) | |
| Flight plan | 85 daily European flights | |
| Aircraft / Engine type | A320/CFM56 engine model | |
| Flight altitude in feet | [FL290, FL410] | |
| Optimization objective | Minimum simple operating cost | Minimum F-ATR20 |

### 5.2 Optimized flight trajectories

We compare the F-ATR20 values of cost-optimized (red) and climate-optimized (green) trajectories in Figure 12. Cost-optimized trajectories are characterized by higher flight altitudes to maximize aerodynamic efficiency, which is similar to what was described in section 4.2. On the other hand, climate-optimized trajectories considering non-$CO_2$ effects fly at lower altitudes at most of the locations to reduce the impact of the total $NO_x$, $H_2O$, and contrails.

Table 5 summarizes the flight characteristics. Compared to the cost-optimized flights, the climate-optimized trajectories (ignoring economic costs while only minimizing climate effects) tend to increase fuel consumption by 17% and $NO_x$ emissions by 25%. On the other hand, the total F-ATR20 is reduced by 51% driven by the contrails effect (-89%), followed by the combined $CH_4$ and PMO impact (-41%). The impact of $CO_2$ and $H_2O$ is characterized by lower orders of magnitudes than the impacts from $NO_x$ emissions and contrails; therefore, they are not crucial properties during the optimization process, but they

are affected by changes due to higher fuel consumption (causing higher $CO_2$ impact (+17%)) and lower mean flight altitudes (leading to lower $H_2O$ impact (-33%)).

Furthermore, one can observe that the contribution of $CO_2$ to the overall climate impact is relatively low compared to the non-$CO_2$ effects. This could be caused by the choice of the physical climate metric and the radiation scheme used to develop the

original CCFs and the following aCCFs characteristics. While ongoing research investigates how to best define an adequate

climate metric reflecting short-term and long-term effects to a certain extent, we expect to develop a better understanding with

further diagnosis.

**Table 5 Daily sum of flight characteristics over the cost-optimized and the climate-optimized trajectories on December 18th 2015.**

| Parameters | Cost-optimized | Climate-optimized | Diff % |
|---|---|---|---|
| Fuel consumption [Tons] | 337.5 | 394.9 | +17.0 |
| $NO_x$ emissions [Tons] | 3.600 | 4.497 | +24.9 |
| Flight time [hrs] | 157.2 | 159.4 | +1.3 |
| Flight distance [km] | 133862 | 137392 | +2.6 |
| Contrail distance [km] | 41032.2 | 30074.1 | -26.7 |
| Cost [thousand EUROs] | 596.9 | 632.1 | +5.9 |
| F-ATR20 $CO_2$ [K] | 2.373e-09 | 2.777e-09 | +17.0 |
| F-ATR20 $H_2O$ [K] | 2.861e-09 | 1.910e-09 | -33.2 |
| F-ATR20 $NO_x$ - $O_3$ [K] | 1.050e-07 | 9.852e-08 | -6.2 |
| F-ATR20 $NO_x$ - $CH_4$+PMO [K] | -2.058e-08 | -2.907e-08 | -41.3 |
| F-ATR20 contrails [K] | 7.209e-08 | 7.644e-09 | -89.4 |
| F-ATR20 total [K] | 1.618e-07 | 7.908e-08 | -51.1 |

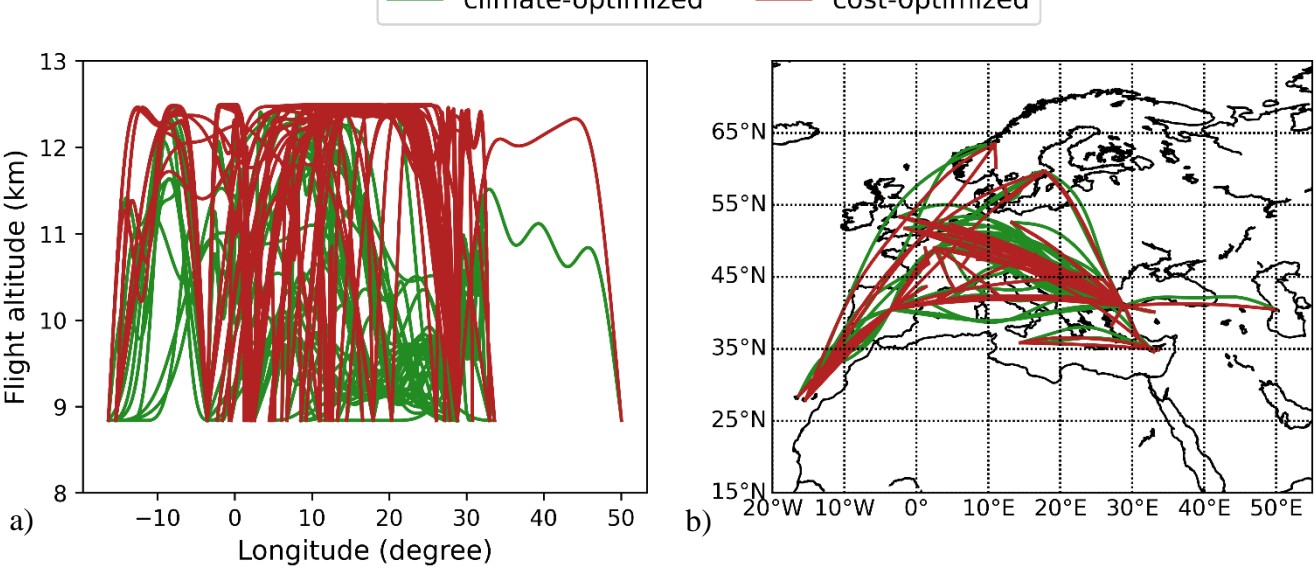

**Figure 12 Comparison of a) vertical shift and b) lateral shift between cost-optimized (red) and climate-optimized (green) trajectories on December 18th 2015.**

## 6    Discussions

This research implements a consistent set of prototype algorithmic climate change functions as the submodel ACCF 1.0 of

EMAC, enabling quantification of aviation emission climate effects. The demonstration simulations confirm that the developed





aCCFs can predict the characteristic patterns of ATR20 from $H_2O$, $NO_x$-induced $O_3$, and contrail-cirrus. The $NO_x$-induced $CH_4$ pattern shows a slight discrepancy in terms of latitudinal variabilities when compared to previous studies (Grewe and Stenke, 2008, Frömming et al., 2012, Köhler et al., 2013). As the total $NO_x$ aCCF is dominated by the positive $O_3$, we expect that the combination of $O_3$ and $CH_4$ captures the feature of aviation $NO_x$ adequately. Further development of $CH_4$ aCCF formula is required to address the latitudinal discrepancy.

Furthermore, the ACCF submodel has been implemented in a comprehensive tagging chemistry simulation chain to evaluate mitigation gains because of modified aviation emissions. By coupling the ACCF submodel with the AirTraf submodel, $NO_x$ emissions are calculated from cost-optimized flights and climate-optimized flights considering only the $NO_x$ induced $O_3$ effect. The $NO_x$ emissions are then fed into the tagging chemistry scheme to estimate the resulting RF due to changes in $O_3$ mixing ratios. The results confirmed that the climate-optimized trajectories reduce the RF of $O_3$ by 2% compared to the cost-optimized

flights.

Nevertheless, the aCCFs 1.0 used in this study represent a prototype formulation and face different aspects of uncertainties. They are based on simulations performed for the North Atlantic Flight Corridor during summer and winter. Additionally, physical climate metrics can be defined on time horizons, e.g., 20, 50, and 100 years. We would like to note here that the development of the aCCFs is an ongoing research activity and an expansion of their geographic scope and seasonal

representativeness is under investigation.

### 6.1 Climate metrics conversion

Regarding the physical climate metric used in this study, the aCCFs formulas in section 3 calculate the average temperature response over 20 years for a pulse emission (P-ATR20). Based on the P-ATR20, it is possible to obtain different physical climate metrics for any other emission scenario by applying a climate response model, e.g., AirClim. Though the flexibility of

the ACCF namelist setup allows the user to convert the climate metrics, the metric selection involves different factors, e.g., the perspective question (Fuglestvedt et al., 2010; Grewe and Dahlmann, 2015). We want to stress that we consider it essential that any optimization study carefully defines the physical climate metric used, the type of strategic decision envisaged, constraints given, and assumptions on policy and regulations accepted. For instance, one should identify the application scenario (or the perspective question) as the specific application scenario is critical for defining the adequate reference, the

physical climate metric, and the emission scenario. A pulse emission would compare the future climate impact in a given year. A future emission scenario would compare the effect of varying emissions over a period in the future. From the perspective question, an adequate climate indicator and time horizon can then be deduced.

### 6.2 Uncertainties of contrail aCCFs

The contrail aCCFs face different sources of uncertainties. Here we discuss two aspects: the sensitivity to the meteorological

input and the uncertainties in the scientific understanding of climate science which is moderate or low (see, e.g., Lee et al., 2021).





The characteristics of aCCFs are sensitive to the input of meteorological conditions. For instance, the potcov varies strongly with the local atmospheric temperature and relative humidity over ice, which again has a dependency on the specific models (e.g., an Earth system climate model vs. a weather forecast model with higher resolution). While comparing the temperature field calculated from the EMAC model on December 18th 2015 nudged towards the ECMWF reanalysis data (ERA-Interim) with the original ERA-Interim datasets at three pressure levels of 200 hPa, 250 hPa, and 300 hPa, we observed that the temperature calculated from the EMAC model is on average 3 K lower than the reanalysis data. This temperature difference affects the predicted potcov and the calculated contrail-cirrus aCCF (see Eq. (6)). Figure 13 shows a comparison between the values of F-ATR20 calculated from contrail-cirrus aCCF on December 18th 2015 at 250 hPa. Figure 13 (a) shows the geographical pattern using the original EMAC temperature, and Figure 13 (b) shows the geographical pattern when artificially correcting the 3 K temperature bias from the EMAC temperature. Two effects are observed: 1) the areas where the contrails might form are reduced for a warmer temperature; 2) the maximum value of ATR20 increases indicating a more substantial warming effect. From this preliminary analysis, we could see that the uncertainties related to the inputs of aCCFs play an essential role in the robustness of the aCCFs results.

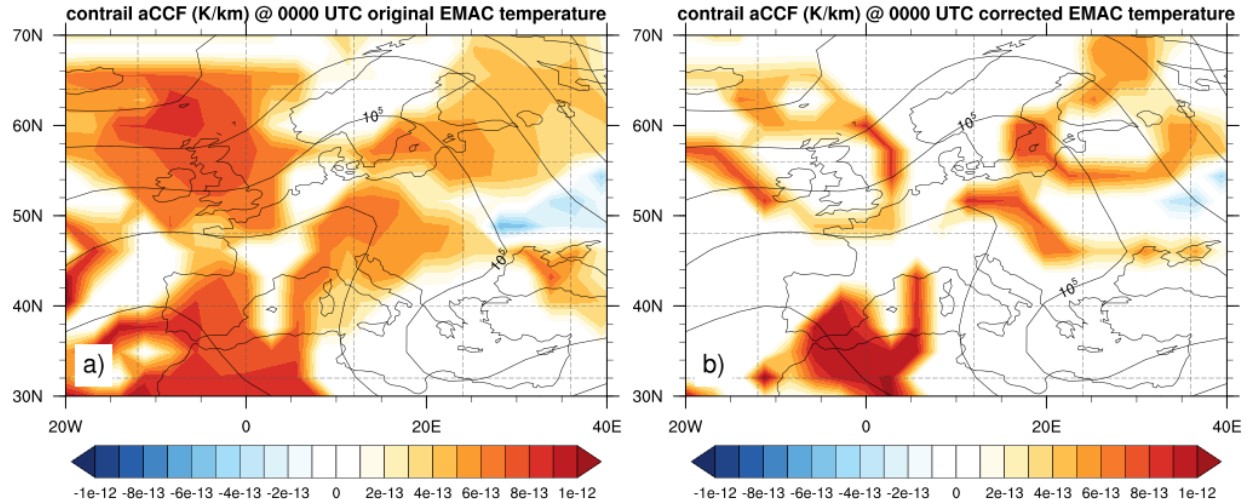

**Figure 13 Geographical distribution of contrail aCCF in K/km on 18th December 2015 at 250 hPa for a) the original EMAC temperature and b) the bias-corrected EMAC temperature.**

### 6.3 Ongoing research on the robustness of aCCFs

While the non-$CO_2$ effects, especially the aviation-induced contrail-cirrus effects, play essential roles in aviation's climate impact, the related uncertainties still need to be resolved. The uncertainties of contrail-cirrus climate impact are subject to different aspects, including the natural variability of the atmosphere and modelling uncertainties. Both uncertainties propagate to contrail-cirrus aCCF. The comprehensive numerical simulations of CCFs were established for weather situations in the North Atlantic Flight Corridor covering the airspace between Europe and the USA. The climatological pattern in vertical and



latitudinal variability matches other studies (Dahlmann et al., 2016). For trajectory optimization, it is essential to understand
how these CCFs deviate in different geographical regions and what that means for climate-optimized flights.

The prototype aCCFs 1.0 still experiences uncertainties in the quantitative estimates in weather forecast and climate impact
prediction. A concept toward robust aCCFs is under development, which will additionally integrate information about
uncertainties arising from low-level understanding of climate science (Matthes et al., 2022, in reparation). This robust aCCFs
will rely on a set of aCCFs that consider educated guess estimates of individual climate impacts. The basis of this educated
guess can be, e.g., the conservative estimates of the individual RF (see Lee et al. 2021). Additionally, the second set of aCCFs
will be provided to perform individual risk analyses originating from different sources of uncertainty. This will be done by
quantitatively estimating the error if a lower or higher climate impact is assumed. With that, we add up to low or high range
aCCFs estimates, respectively. This concept of robust aCCFs can be applied in aircraft trajectory optimization studies with
EMAC/AirTraf. The corresponding experiment design would rely on one reference optimization using the educated guess
aCCFs and sensitivity optimization experiments using the low- or high-range aCCFs estimates. A robust trajectory would be
characterized by not losing overall benefits (mitigation gains) even if lower or upper estimates of aCCFs are applied.
Technically, this could be solved by calling the ACCF submodel several times within the same simulation, using the range of
different aCCFs estimates.

## 7   Conclusions

We developed the submodel ACCF 1.0 of the chemistry-climate model EMAC to estimate the climate impact of aviation
emissions in the flight corridor of the northern hemisphere representing an implementation of aCCFs 1.0 formulas. The
submodel ACCF 1.0 was developed according to the MESSy standard and was thoroughly presented in this paper. This
submodel calculates aviation's climate impact of $CO_2$ emissions and non-$CO_2$ effects, such as from $NO_x$-induced $O_3$, $NO_x$-
induced $CH_4$ (including PMO), $H_2O$, and contrail-cirrus based on a consistent set of aCCFs. The mathematical formulation of
the individual prototype aCCFs 1.0 is provided.

The climatological profile of the $NO_x$-induced effect on ozone ($O_3$ aCCF) shows that the warming effects of $NO_x$-induced $O_3$
increase with the altitude between 150-300 hPa and towards lower latitudes. While the climatological distribution of $H_2O$
aCCF shows that the warming effect of $H_2O$ increases towards higher altitudes or latitudes. By comparing to literature, we
conclude that the vertical and latitudinal structure within the flight corridor of the northern hemisphere of the $NO_x$ induced $O_3$
and $H_2O$ are well represented by the aCCFs.

The $NO_x$-induced effect on methane ($CH_4$ aCCF) shows that cooling effects increase towards lower altitudes and higher
latitudes. Although the latitudinal variation of $CH_4$ aCCFs is less pronounced than for other species, it is somewhat of the
opposite tendency to the literature. Since the absolute value of $CH_4$ aCCF is mostly overcompensated by the $O_3$ aCCF, the
total $NO_x$ aCCF could still capture the vertical and latitudinal variability of the overall $NO_x$ effects.

For the contrail-cirrus aCCF, the climatological pattern follows the potential contrail coverage. The calculated F-ATR20 value
also matches the literature, except that contrail-cirrus aCCF generates values at low altitudes where contrails are not expected



to be formed. This might be related to the threshold of temperature and humidity used for calculating the potential contrail coverage and the temperature bias present in the EMAC model.

Using the tagging chemistry approach, we were able to show that climate-optimized trajectories based on $O_3$ aCCF indeed
reduce the radiative forcing contribution from aviation $NO_x$ induced $O_3$ compared to the cost-optimized trajectories.

Finally, the trajectory optimization results confirm that the total F-ATR20 of climate-optimized flights is about 51% lower than the cost-optimized flights, with the largest contribution from contrail-cirrus.

**Code availability**

ACCF 1.0 has been published for the first time as a submodel of the Modular Earth System Submodel System (MESSy) since
version 2.53. MESSy is continuously further developed and applied by a consortium of institutions. The usage of MESSy and access to the source code are licensed to all affiliates of institutions members of the MESSy Consortium by signing the MESSy Memorandum of Understanding. More information can be found on the MESSy Consortium Website (http://www.messy-interface.org ). The version presented here corresponds to ACCF 1.0. The status information for ACCF will be available on the website.

**Supplement**

The supplement related to this paper includes the development of contrail-cirrus aCCF and the user manual for the ACCF submodel setup.

**Author contributions**

FY and VG designed the submodel ACCF V1.0. FY implemented the coupling of ACCF V1.0 with the Modular Earth
Submodel System (MESSy). VG, SM, KD, EK, and KS developed the algorithmic Climate Change Functions (aCCFs). KD calculated the metric conversion factor. CF calculated the Climate Change Functions (CCFs). HY and VG designed the submodel AirTraf V2.0. BL and FL provided the traffic sample for this study. PR, SD, SM, and PP contributed to the discussions. FY and FC performed the simulations and analysed the results presented in this paper.

**Competing interests**

The authors declare that they have no conflict of interest.

**Acknowledgment**

The current study has been supported by the previous ATM4E project and the current FlyATM4E project. Both projects have received funding from the SESAR Joint Undertaking under grant agreements No. 699395 (ATM4E) and No. 891317 (FlyATM4E) under European Union's Horizon 2020 research and innovation program.





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



**Appendix A list of EMAC submodels used in the chemistry simulation**

**Table A.1 Summary of MESSy submodels used in the chemistry simulation.**

| Submodel | Purpose | Reference |
|---|---|---|
| AEROPT | Aerosol optical properties for the radiation scheme | Dietmüller et al., 2016 |
| ACCF 1.0 | Climate impact of aviation emissions and contrails calculation | Section 3 of this article |
| AIRTRAF 2.0 | Air traffic simulation | Yamashita et al., 2020 |
| CH4 1.0 | Simple methane chemistry | Winterstein and Jöckel, 2021 |
| CLOUD | Standard ECHAM5 cloud microphysics calculation | Roeckner et al., 2006 |
| CLOUDOPT | Cloud optical properties calculation for the radiation scheme | Dietmüller et al., 2016 |
| CVTRANS | Calculates the transport of tracers due to convection | Tost, 2006 |
| CONVECT | Convection process calculation | Tost et al., 2006b |
| CONTRAIL | Contrail potential coverage calculation | Supplement of Grewe et al., 2014a; Yin et al., 2018 |
| DDPE | Dry deposition of gas phase and aerosol tracers | Kerkweg et al., 2006a |
| E5VDIFF | ECHAM5 vertical diffusion and land-atmosphere exchange | Jöckel et al., 2010 |
| GWAVE | Gravity waves calculation | Jöckel et al., 2010 |
| JVAL | Photolysis rates | Sander et al., 2014 |
| LNOX | Lighting $NO_x$ production | Tost et al., 2007 |
| MSBM | Multi-phase stratospheric box model calculates the heterogeneous reaction rates on polar stratospheric cloud particles and stratospheric background aerosols | Jöckel et al., 2010 |
| MECCA | Calculates tropospheric and stratospheric chemistry | Sander et al., 2005 |
| O3ORIG | To trace the origin of ozone | Grewe, 2006 |
| OFFEMIS | Prescribed emissions of trace gases and aerosols | Kerkweg et al., 2006b |
| ONEMIS | Online calculated emissions of trace gases and aerosols | Kerkweg et al., 2006b |
| ORBIT | Earth orbit calculation for solar zenith angle, etc. | Dietmüller et al., 2016 |
| RAD | Simulates the radiative flux | Dietmüller et al., 2016 |
| SCAV | Simulates the process of wet deposition and liquid phase chemistry. | Tost et al., 2006a |
| SCAL | Simple calculations with channel objects to separate the AirTraf ozone from other ozone sources | Jöckel et al., 2010 |
| SEDI | Sedimentation of aerosol particles | Kerkweg et al., 2006a |
| SURFACE | Calculates the surface temperature | Jöckel et al., 2010 |





| TAGGING 1.0 | Tag the emissions contributions to concentrations | Grewe et al., 2017a |
| TNUDGE | Tracer nudging | Kerkweg et al., 2006b |
| TROPOP | Tropopause and other diagnosis | Jöckel et al., 2006 |
| VISO | Vertically layered iso-surfaces and maps | Jöckel et al., 2010 |