# Peer review of "Predicting the climate impact of aviation for en-route emissions: The algorithmic climate change function submodel ACCF 1.0 of EMAC 2.53"

_Geoscientific Model Development, 2022_

## Author Response (AR1)

We are grateful to referee #1 for the constructive and encouraging comments on the original version of our manuscript. We took all comments into account and revised the manuscript accordingly. Here are our replies:

- This study presents the ACCF 1.0 to describe the climate impact of aviation emissions. ACCF 1.0 takes the atmospheric conditions as input to calculate the climate impact, mainly through the average temperature response over 20 years (ATR20). The emissions include, (via and), vapour, and contrail-cirrus. The study is valuable as it provides an integrated model to assess the environmental impact of non-$CO_2$ emissions. I have a few comments below:

  **Reply**: We thank referee #1 for these positive comments. We have addressed all the comments as follows.

- The ACCF 1.0 model is based on the aCCF, which was proposed by the earlier project REACT4C2 for researching the climate change caused by emissions. ACCF works as a submodel of the global atmospheric-chemistry model EMAC. What new features/functions are developed should be discussed.

  **Reply**: As the reviewer might have noticed in Figure 1 of this manuscript, there are three stages of work leading to the current version of algorithmic climate change functions (aCCFs by short), which is the core of the ACCF submodel presented in this manuscript.

  *Stage one*: In REACT4C, the original Climate Change Functions (CCFs by short) were developed and implemented for climate optimized flight trajectories, including the effects from $CO_2$, $NO_x$, $H_2O$, and contrail cirrus (Grewe et al., 2014a). The various case studies showed the effectiveness of using CCFs to achieve flight routings with minimum climate impact (Grewe et al. 2014b; Grewe et al. 2017b). The main challenge is that CCFs cannot be directly implemented in trajectory planning tools since generating CCFs requires a heavy computational load.

  This brought the work to the *second stage* to develop algorithmic CCFs in a previous EU project ATM4E (van Manen and Grewe, 2019; Matthes et al., 2017). aCCFs are simplified response models for estimating the climate impact of various aviation emissions (e.g., $CO_2$, $H_2O$ and $NO_x$)/effect (day/night contrails). Though the inputs to aCCFs are the original CCFs, the step from CCFs to aCCFs is innovative and essential, as unlike CCFs, aCCFs can be directly implemented in flight planning tools to obtain climate optimized flight trajectories quickly. The aCCFs for $NO_x$ and $H_2O$ estimating has been well documented by van Manen and Grewe (2019). The approach of developing contrails aCCFs is only made available in the current ACCF v1.0 manuscript as a supplement.

  While continuing to the *third stage* of implementing the aCCFs in the current ACCF V1.0 (FlyATM4E scope), we noticed inconsistencies between the $NO_x$/$H_2O$ aCCFs and contrails aCCFs due to different approaches in calculating radiative forcing and the emission scenarios concerned to develop the response models (Pulse vs. future increasing emissions). For details, please see the discussion in section 2 of this manuscript. Therefore, one significant effort in the current manuscript is to diagnose and correct the previous aCCFs models to be consistent.

  Long story short, the novelty of the current paper is that we, for the first time, publish: 1) the contrail aCCFs development; 2) a consistent set of aCCFs models in terms of fuel scenario, metric, and efficacy for all species.

- The application simulation of existing trajectory is conducted to show the climatology impact. They also used the calculation model to optimize the trajectory, from which they draw the

conclusion that climate-optimized trajectories considering non-$CO_2$ effects fly lower altitudes to reduce the impact of the total $NO_x$, $H_2O$, and contrails. The scalability of the tool for large-scale problems should be discussed.

**Reply**: The application study in section 5 is based on a daily simulation. Accordingly, climate-optimized flights tend to reduce their flight altitudes driven by $NO_x$ and contrail effects. Regarding the tool's scalability, the quality check in section 4 can be seen as a scalability check of aCCFs models themselves. For instance, in section 4.1, we compared the climatological aCCFs with the literature to confirm that the pattern of aCCFs matches the previous studies. Furthermore, in general, $NO_x$'s climate impact decreases at low altitudes and high latitudes due to shorter residence time or inactive atmospheric chemistry process of $NO_x$. The climate impact of $H_2O$ reduces with altitude as well. The effect of contrails is vital in a narrow altitude band following the tropopause height. Following such behavior, we expect the flights with lower climate impact will favor a lower flight altitude than cost-optimal flights. We are performing work based on a year simulation to quantify the mitigation gains regarding climate impact.

- However, in their scenarios, the $CO_2$-related environmental impact is considered to be lower than the non-$CO_2$ impact, which may limit the possible subsequent applications of the ACCF. Maybe discuss futural models, which can provide a comprehensive assessment of climate impact caused by aviation emissions.

**Reply**: We agree that the lower $CO_2$ impact estimated from the aCCFs model requires further diagnosis and new developments if required. During our current work, we performed intensive studies to understand the mechanism, and we suspect the reasons could be in the following aspects:

1) Metrics: when we convert the metrics to 100 years' time horizon instead of 20 years in this paper, we observed an increase in $CO_2$ contribution for the same set of flights.
2) Radiation scheme for developing the original CCFs.
3) Geographical representative: the current research is mainly on European regions. Ongoing research is investigating whether $CO_2$ and non-$CO_2$ relative importance are globally consistent, and this will help us understand better the local representation of aCCFs as well.

We have added one paragraph on page 24 Line 505-508, for this discussion. Please see the track change in the resubmission.

**Reference**

- Grewe, V., Frömming, C., Matthes, S., Brinkop, S., Ponater, M., Dietmüller, S., Jöckel, P., Garny, H., Tsati, E. and Dahlmann, K. (2014a). "Aircraft routing with minimal climate impact: the REACT4C climate cost function modelling approach (V1. 0)." Geoscientific Model Development, 7(1): 175-201. DOI: https://doi.org/10.5194/gmd-7-175-2014.
- Grewe, V., Champougny, T., Matthes, S., Frömming, C., Brinkop, S., Søvde, O. A., Irvine, E. A. and Halscheidt, L. (2014b). "Reduction of the air traffic's contribution to climate change: A REACT4C case study." Atmospheric Environment, 94: 616-625. DOI: https://doi.org/10.1016/j.atmosenv.2014.05.059.
- Grewe, V., Matthes, S., Frömming, C., Brinkop, S., Jöckel, P., Gierens, K., Champougny, T., Fuglestvedt, J., Haslerud, A., Irvine, E. and Shine, K. (2017b). "Feasibility of climate-optimized air traffic routing for trans-Atlantic flights." Environmental Research Letters, 12(3): 034003. DOI: 10.1088/1748-9326/aa5ba0.

o   van Manen, J. and Grewe, V. (2019). "Algorithmic climate change functions for the use in eco-efficient flight planning." Transportation Research Part D: Transport and Environment, 67: 388-405. DOI: https://doi.org/10.1016/j.trd.2018.12.016.

o   Matthes, S., Grewe, V., Dahlmann, K., Frömming, C., Irvine, E., Lim, L., Linke, F., Lührs, B., Owen, B., Shine, K., Stromatas, S., Yamashita, H. and Yin, F. (2017). "A Concept for Multi-Criteria Environmental Assessment of Aircraft Trajectories." Aerospace, 4(3): 42.

We are grateful to referee #2 for the constructive comments on the original submission. We took those comments into account and revised the manuscript accordingly. Furthermore, we observe that referee #2 misunderstands our work in several places, particularly the model development and applicability. We hope with the point-to-point replies below, those doubts will be further clarified.

- The paper deals with an interesting approach to determining the climate impact on air traffic. The abstract lacks motivation, results, and applicability. The first sentence of the abstract has no content. Already in the abstract, there are numerous unexplained abbreviations.

  **Reply**: Thank you for the feedback. We revised the abstract intensively to include motivation, the results, and the recommendations in terms of applicability. We also added the full names of abbreviations whenever required. Please refer to the track change in the revision.

- In the introduction, the motivation is based on a 4-year-old prediction. This should be made acute.

  **Reply**: We agree that the Airbus prediction was made in 2018 based on the pre-Covid scenario. Meanwhile, the authors are aware of other predictions considering Covid-impact. For instance, as pointed out in our introduction (please see page 2 Line 42-43), the ICAO post-covid-19 forecasts show that the revenue passenger kilometer (RSK) would recover in the long term (e.g., an average annual growth of 3.6% with a low and high possibility of (2.9%, 4.2%) over 30 years from 2018 to 2050). Furthermore, the other recent IATA prediction shows an average annual ASK growth rate of 3.3% by 2040. Given that prediction mostly contains uncertainties, the airbus 4.4% still falls that similar order and would be a good motivation for mitigating aviation's climate impact.

  Nevertheless, we appreciate the reviewer's critical attitude. To make it more explicit, we revised the text by adding: "An example from the recent ICAO post-COVID forecast shows that the Revenue Passenger- Kilometres (RPK) is expected to grow at an annual average rate of 3.6% with a low and high range between 2.9% and 4.2% over the next three decades from 2018 to 2050". (please see the track change page 2 Line 44-Line45)

  Reference:
  - https://www.icao.int/sustainability/Pages/Post-Covid-Forecasts-Scenarios.aspx
  - IATA, Global Outlook for Air Transport, 2022. (https://www.iata.org/en/iata-repository/publications/economic-reports/global-outlook-for-air-transport---december-2022/).

- The state of the art is completely missing. Instead, we find a paragraph with far too many self-citations, which summarises preliminary views of the authorship.

  **Reply**: We updated the reference list to include more recent studies relating to aviation's $NO_x$ impact (Lund et al., 2017; Szopa et al., 2021; Terrenoire, et al., 2022). Please see also the highlights in the revision page 2. Overall, the climate impact of $NO_x$ emissions consists of ozone formation (warming effect), and methane depletion (cooling effect) remains. One might expect a change in the severity of aviation $NO_x$'s impact when the background concentrations, for instance, methane, change in the future (Terrenoire, et al. 2022). But this is beyond the scope of the aCCFs model as we mainly consider the current scenario, which is of great urgency.

  Furthermore, we include the recent research related to Covid impact. For instance, Voigt et al. (2022) conducted a measurement campaign to investigate atmospheric concentration changes. The authors observed a significant reduction in $NO_x$ at cruise altitudes, contrail

coverage, and the resulting radiative forcing. Gettelman et al. (2021) show that the effect of COVID-19 reductions in flights reduces contrail formation. However, due to spatial and seasonal variability of contrail radiative forcing, the annual mean contrail effective radiative forcing shows no significant changes.

All the updated references are listed below and highlighted in the revision with yellow color coding.

Newly added reference:
- o Lund, M. T., Aamaas, B., Berntsen, T., Bock, L., Burkhardt, U., Fuglestvedt, J. S. and Shine, K. P. (2017). "Emission metrics for quantifying regional climate impacts of aviation." Earth Syst. Dynam., 8(3): 547-563. DOI: 10.5194/esd-8-547-2017.
- o Szopa, S., Naik, V., Adhikary, B., Artaxo, P., Berntsen, T., Collins, W. D., Fuzzi, S., Gallardo, L., Kiendler-Scharr, A., Z. Klimont, Liao, H., Unger, N. and Zanis, P. (2021). Short-Lived Climate Forcers. In Climate Change 2021: The Physical Science Basis. Contribution of Working Group I to the Sixth Assessment Report of the Intergovernmental Panel on Climate Change. [V. Masson-Delmotte, P. Zhai, A. Pirani, S. L. Connors, C. Péan, S. Berger, N. Caud, Y. Chen, L. Goldfarb, M. I. Gomis, M. Huang, K. Leitzell, E. Lonnoy, J. B. R. Matthews, T. K. Maycock, T. Waterfield, O. Yelekçi, R. Yu and B. Zhou (eds.)]. Cambridge University Press, Cambridge, United Kingdom and New York, NY, USA, pp.817-922. DOI: https://doi.org/10.1017/9781009157896.008.
- o Terrenoire, E., Hauglustaine, D. A., Cohen, Y., Cozic, A., Valorso, R., Lefèvre, F. and Matthes, S. (2022). "Impact of present and future aircraft NOx and aerosol emissions on atmospheric composition and associated direct radiative forcing of climate." Atmos. Chem. Phys., 22(18): 11987-12023. DOI: 10.5194/acp-22-11987-2022.
- o Gettelman, A., Chen, C. C. and Bardeen, C. G. (2021). "The climate impact of COVID-19-induced contrail changes." Atmos. Chem. Phys., 21(12): 9405-9416. DOI: 10.5194/acp-21-9405-2021.
- o Voigt, C., Lelieveld, J., Schlager, H., Schneider, J., Curtius, J., Meerkötter, R., Sauer, D., Bugliaro, L., Bohn, B., Crowley, J. N., Erbertseder, T., Groß, S., Hahn, V., Li, Q., Mertens, M., Pöhlker, M. L., Pozzer, A., Schumann, U., Tomsche, L., Williams, J., Zahn, A., Andreae, M., Borrmann, S., Bräuer, T., Dörich, R., Dörnbrack, A., Edtbauer, A., Ernle, L., Fischer, H., Giez, A., Granzin, M., Grewe, V., Harder, H., Heinritzi, M., Holanda, B. A., Jöckel, P., Kaiser, K., Krüger, O. O., Lucke, J., Marsing, A., Martin, A., Matthes, S., Pöhlker, C., Pöschl, U., Reifenberg, S., Ringsdorf, A., Scheibe, M., Tadic, I., Zauner-Wieczorek, M., Henke, R. and Rapp, M. (2022). "Cleaner Skies during the COVID-19 Lockdown." Bulletin of the American Meteorological Society, 103(8): E1796-E1827. DOI: 10.1175/bams-d-21-0012.1.

- The work is based on Climate costs functions CCF, which is not comprehensibly derived in any of the sources mentioned. The errors of the CCF are not discussed. The transferability to other time periods is very questionable and is not discussed.

**Reply**: The Climate Cost Functions are published in Grewe et al. (2014a), which was also used as one of our references. There are extensive descriptions of the CCFs development. The verification of CCFs has been performed with a thorough comparison to other modelling results in terms of the temporal evolution of species, chemistry, and properties of contrails, metrics, and the validity of the resulting flight trajectories. These processes are also documented in Grewe et al. (2014a).

Furthermore, in section 4.1 of Grewe et al. (2017b), the authors summarized various sources of certainties. For instance, the prediction of persistent contrail location and the contrail characteristics link to the micro-physical process, the aircraft/fuel characteristics, etc. Addressing those uncertainties within such a complex model chain is not trivial and is beyond the scope of this research. Nevertheless, in different studies, we investigated the impact of uncertainties related to the weather forecast by implementing ensemble weather data, on the climate optimized flight trajectories using aCCFs model (Simorgh et al., 2022). We expect, with the ongoing research through Monte-Carlo simulations, we will be able to investigate further the uncertainties of CCFs in the future.

Regarding the transferability to other time periods: we feel like the reviewer has misunderstood the approach. Here, we try to elaborate further. Irvine et al. (2013) identified 5 winter weather patterns and 3 summer weather patterns, based on which the relation between weather data and the climate impact of aviation emissions was derived and used to develop the original CCFs model as described in Grewe et al. (2014a). van Manen and Grewe (2019) used the CCFs datasets to analyze the link of weather data to different aviation's climate effects, e.g., $NO_x$-induced $O_3$, $NO_x$-induced $CH_4$ and $H_2O$. Accordingly, the aCCFs were developed. Though the aCCFs have been developed based on the CCFs data, the formality has been generalized beyond the specific days per weather pattern in CCFs. For instance, Yin et al. (2018) and Rao et al. (2022) performed case studies of implementing the $NO_x$ aCCFs on arbitrary day weather conditions concerning European flights. By coupling with the chemistry process, the simulations confirmed the effectiveness of using $O_3$ aCCF model for climate optimized trajectory to reduce the RF of aviation $NO_x$-induced $O_3$. Therefore, time transferability to a different days per weather situation is not a major issue in the authors' opinion.

On the other hand, the authors are aware that the original CCFs have been developed for the North Atlantic Flight Corridor during summer and winter. This also suggests that using aCCFs at other locations or seasons should be carefully evaluated. We included this in the discussion section; see Lines 509-513 on page 24. Further research is ongoing to expand the model's geographical location and time coverage. We expect that ACCF 1.0 will be updated in the future accordingly.

Reference:

- o Grewe, V., Frömming, C., Matthes, S., Brinkop, S., Ponater, M., Dietmüller, S., Jöckel, P., Garny, H., Tsati, E. and Dahlmann, K. (2014a). "Aircraft routing with minimal climate impact: the REACT4C climate cost function modelling approach (V1. 0)." Geoscientific Model Development, 7(1): 175-201. DOI: https://doi.org/10.5194/gmd-7-175-2014.
- o Grewe, V., Matthes, S., Frömming, C., Brinkop, S., Jöckel, P., Gierens, K., Champougny, T., Fuglestvedt, J., Haslerud, A., Irvine, E. and Shine, K. (2017b). "Feasibility of climate-optimized air traffic routing for trans-Atlantic flights." Environmental Research Letters, 12(3): 034003. DOI: 10.1088/1748-9326/aa5ba0.
- o Simorgh, A., Soler, M., González-Arribas, D., Linke, F., Lührs, B., Meuser, M. M., Dietmüller, S., Matthes, S., Yamashita, H., Yin, F., Castino, F., Grewe, V., and Baumann, S.: Robust 4D Climate Optimal Flight Planning in Structured Airspace using Parallelized Simulation on GPUs: ROOST V1.0, EGUsphere [preprint], https://doi.org/10.5194/egusphere-2022-1010, 2022.

- Irvine, E. A., Hoskins, B. J., Shine, K. P., Lunnon, R. W. and Froemming, C. (2013). "Characterizing North Atlantic weather patterns for climate-optimal aircraft routing." Meteorological Applications, 20(1): 80-93. DOI: 10.1002/met.1291.
- Yin, F., Grewe, V., van Manen, J., Matthes, S., Yamashita, H., Linke, F., and Lührs, B., "Verification of the ozone algorithmic climate change functions for predicting the short-term NOx effects from aviation en-route", ICRAT conference, 2018.
- Rao, P., Yin, F., Grewe, V., Yamashita, H., Jöckel, P., Matthes, S., Mertens, M. and Frömming, C. (2022). "Case Study for Testing the Validity of NOx-Ozone Algorithmic Climate Change Functions for Optimising Flight Trajectories." Aerospace, 9(5): 231.
- van Manen, J. and Grewe, V. (2019). "Algorithmic climate change functions for the use in eco-efficient flight planning." Transportation Research Part D: Transport and Environment, 67: 388-405. DOI: https://doi.org/10.1016/j.trd.2018.12.016.

- The scientific amount of Figure 1 to the paper is not made clear.

  **Reply**: The roadmap toward the ACCF 1.0 model involves multiple stages of work originating from different research projects. The development of CCFs and aCCFs are closely related but still different processes. With Figure 1, we intend to guide the reader through the process and illustrate distinctions of the model development processes between CCFs, the first version aCCFs, and the implementation of aCCFs in ACCF 1.0. Please also see the explanations in the previous point above.

  Also, with this figure, we intend to demonstrate the main contributions of the current research. While the original CCFs and the first aCCFs model have been developed and published in the previous research, the approach of developing contrails aCCFs is made available in the current ACCF v1.0 manuscript as a supplement. Furthermore, one main effort of this research is to derive a consistent set of aCCFs models and evaluate the quality of the aCCFs.

  We revised the paper to make the purpose of figure 1 clear. Please see the revision page 4, Line 114-Line 126. Figure 1 is also updated.

- Equations 1 and 2 were copied from Manen and Grewe and should be properly cited.

  **Reply**: While the format of Eqn. 1 and 2 follows the main thought of van Manen and Grewe (2019), the coefficients in this paper differ from the van Manen and Grewe study. The reason is that we intend to provide a set of formulas for pulse emission scenarios, which can be used as the basis to derive other metrics like future emission scenarios. Also, with the updates, a more consistent conversion from radiative forcing to average temperature response (ATR) is employed (explained in Lines 143-149, page 6).

  We added such an explanation in Lines 222-226 on page 9. Please refer to the track change in revision.

  Reference:

  - van Manen, J. and Grewe, V. (2019). "Algorithmic climate change functions for the use in eco-efficient flight planning." Transportation Research Part D: Transport and Environment, 67: 388-405. DOI: https://doi.org/10.1016/j.trd.2018.12.016.

- The constant factor 0.0151 K/W/m2 in line 244 should be critically questioned, and its error should be critically discussed.

  **Reply**: The constant factor of 0.0151 $K/W/m^2$ for converting radiative forcing to average temperature response is obtained using the climate response model, AirClim (Grewe and Stenke,

2008; Dahlmann et al., 2016), considering a consistent set of scenarios as to other specifies in this research. We apply a consistent set of global emission inventory for a given scenario, for which the RF and ATR20 are calculated. The ratio between RF and ATR20 is then derived as 0.0151 K/W/m², hence used as a conversion factor. In general, the conversion between radiative forcing to average temperature response follows linear relation; therefore, we consider using a constant factor a reasonable approach.

Please see page 6 Line 143-151 and page 13 Line 305-308 of the manuscript for revision.

Reference:

- o Grewe, V. and Stenke, A. (2008). "AirClim: an efficient tool for climate evaluation of aircraft technology." Atmospheric Chemistry and Physics, 8(16): 4621-4639. DOI: 10.5194/acp-8-4621-2008.
- o Dahlmann, K., Grewe, V., Frömming, C. and Burkhardt, U. (2016). "Can we reliably assess climate mitigation options for air traffic scenarios despite large uncertainties in atmospheric processes?" Transportation Research Part D: Transport and Environment, 46: 40-55. DOI: https://doi.org/10.1016/j.trd.2016.03.006.

- The sole distinction between day and night is not sufficient in the context of the Contrail RF and ignores cooling effects during sunrise and sunset.

  **Reply**: The authors are fully aware of the day and night difference for contrail RF. For instance, Figure 6 of Frömming et al. (2021) shows the instantaneous RF at the top of the atmosphere of individual contrail concerning location time (night, twilight, and day). It is observable that night contrail causes positive RF; through twilight and day, the RF could be positive or negative. Such differences are considered in the derivation of contrail aCCFs. As can be seen in Eqn. (6) the RF of nighttime contrail is a function of temperature (>201 K) with positive RF expected. And the daytime contrail aCCF, in Eqn. (8), depending on the outgoing longwave radiation (OLR) value. One expects a negative daytime contrail RF for OLR smaller than -193 W/m². Figure 6 of this manuscript also demonstrates that for a contrail formed at 12:00 UTC in Winter Europe, one expects cooling and warming effects. For a contrail occurring at midnight 00:00 UTC, pure warming effects are expected.

  Reference:

  - o Frömming, C., Grewe, V., Brinkop, S., Jöckel, P., Haslerud, A. S., Rosanka, S., van Manen, J. and Matthes, S. (2021). "Influence of the actual weather situation on aviation climate effects: The REACT4C Climate Change Functions." Atmospheric Chemistry and Physics, 21: 9151-9172. DOI: https://doi.org/10.5194/acp-21-9151-2021.

- The extreme heterogeneity of the contrail CCFs in Figure 6 supports the assumption that the developed CCFs are extremely weather-dependent and thus not applicable to other time periods.

  **Reply**: For this question, we think this is a misunderstanding of the aCCFs approach. Therefore, we would like to refer to one of the previous reply (bullet #4) above relating to the methodology of aCCFs and the transferability to other days. To be short, we analyzed the weather situation from CCFs weather patterns to gain an understanding; however, the derivation is done on a general basis for covering the whole range of weather patterns.

- Please explain why the effectiveness in line 285 is not included in the CCF and derive the uncertainty of the effectiveness.

**Reply**: We are not sure if we understood the reviewer's comments, as, in line 285, it is on efficacy description instead of "effectiveness". In line with the efficacy thought, we agree it is a good idea to include this in the original CCFs development; however, to our knowledge, the scope was more on deriving the CCFs as the efficacy is often a constant multiplier and can be obtained from other studies (e.g., Lee et al., 2021).

- In Figure 9, your definition of a cost-optimal and a climate-optimal trajectory is absolutely necessary to understand the procedure.

  **Reply**: Cost-optimal aircraft trajectory minimizes the simple operating cost of the flight, consisting of the cost related to flight time and fuel consumption, as defined in Eqn. (10) of the manuscript, climate-optimal aircraft trajectory minimizes the ATR20 value from only the $NO_x$-induced $O_3$ effect. The explanation can be found on page 19 Lines 415-419.

  To make it more explicit, we now add the explanation in the caption of figure 9. Please see the track change in the revision.

- The dents and ripples in the optimised trajectories in Figures 10 and 11 should definitely be explained and critically questioned.

  **Reply**: The optimized flight trajectories are presented in Figure 10 a) and b). Figure 11 is one mixing ratio of $NO_x$ and $O_3$. Regarding the dents and ripples in optimized trajectories of Figure 10 a) and b), there are multifold reasons:

  - The flight trajectory is defined per flight and in total 85 flights are considered here in this study. In the actual flight schedule, they depart at different time steps, which is not visible in these 2D figures. As a result, all the flights seemed to overlap with the large variability shown.

  - Furthermore, the respective optimized flight trajectories for short and long flights are projected on longitude vs. latitude, which causes the dents and ripples, as shown in Figure 10 (a).

  - Per climate-optimized flight, the optimizer tends to detour a flight at a specific location where the $NO_x$-induced $O_3$ effect is strong. This causes altitude variability for climate-optimized flights. Such behavior is typical when one addresses the climate impact of non-$CO_2$ effects in trajectory optimization (e.g., Figure 1 of Matthes et el., 2020). In general, cost-optimized flights tend to fly at the ceiling altitude for low fuel consumption and hence low operational cost. But climate-optimized flights vary per location. See page 20 Lines 430-432 of this manuscript.

  Reference:
  - Matthes, S.; Lührs, B.; Dahlmann, K.; Grewe, V.; Linke, F.; Yin, F.; Klingaman, E.; Shine, K.P (2020). "Climate-Optimized Trajectories and Robust Mitigation Potential: Flying ATM4E". Aerospace, 7, 156. https://doi.org/10.3390/aerospace7110156

- All results and assumptions should have been critically questioned and discussed in the conclusions at the latest. An error analysis of such a strongly empirically driven model is absolutely necessary.

  **Reply**: We agree with the reviewer that the assumptions and uncertainties are critical to the work results and the model's performance. Next to the research done so far, in discussion section 6, we discussed the findings of this study intensively with a primary focus on further work to address the uncertainties of aCCFs modelling approach, for instance, selection of

climate metrics, uncertainties of persistent contrails prediction and the implication to aCCFs, the robustness of aCCFs (e.g., the impact of weather forecast uncertainties, the aircraft engine technology assumption, the prediction of emissions, etc.), and the representative of $CO_2$ vs. non-$CO_2$ effects. Since it is a complex modelling process, uncertainty could occur at each step. There is ongoing work to address different aspects of the uncertainties. We feel it is a step by step approach, and in this paper, we like to document the aCCFs consistently, which also serves as a basis for future work.